# AutoBayes: Automated Bayesian Graph Exploration for Nuisance-Robust Inference

## Abstract

Learning data representations that capture task-related features, but are invariant to nuisance variations[1] remains a key challenge in machine learning. We introduce an automated Bayesian inference framework, called AutoBayes, that explores different graphical models linking classifier, encoder, decoder, estimator and adversarial network blocks to optimize nuisance-invariant machine learning pipelines. AutoBayes also enables learning disentangled representations, where the latent variable is split into multiple pieces to impose various relationships with the nuisance variation and task labels. We benchmark the framework on several public datasets, and provide analysis of its capability for subject-transfer learning with/without variational modeling and adversarial training. We demonstrate a significant performance improvement with ensemble learning across explored graphical models.

## 1 Introduction

The great advancement of deep learning techniques based on deep neural networks (DNN) has enabled more practical design of human-machine interfaces (HMI) through the analysis of the user's physiological data (Faust et al., 2018), such as electroencephalogram (EEG) (Lawhern et al., 2018) and electromyogram (EMG) (Atzori et al., 2016). However, such biosignals are highly prone to variation depending on the biological states of each subject (Christoforou et al., 2010). Hence, frequent calibration is often required in typical HMI systems.

Toward resolving this issue, subject-invariant methods (Özdenizci et al., 2019b), employing adversarial training (Makhzani et al., 2015; Lample et al., 2017; Creswell et al., 2017) with the Conditional Variational AutoEncoder (A-CVAE) (Louizos et al., 2015; Sohn et al., 2015) shown in Fig. 1(b), have emerged to reduce user calibration for realizing successful HMI systems. Compared to a standard DNN classifier $\mathcal{C}$ in Fig. 1(a), integrating additional functional blocks for encoder $\mathcal{E}$, nuisance-conditional decoder $\mathcal{D}$, and adversary $\mathcal{A}$ networks offers excellent subject-invariant performance. The DNN structure may be potentially extended with more functional blocks and more latent nodes as shown in Fig. 1(c). However, such a DNN architecture design may rely on human effort and insight to determine the block connectivity of DNNs. Automation of hyperparameter and architecture exploration in the context of AutoML (Ashok et al., 2017; Brock et al., 2017; Cai et al., 2017; He et al., 2018; Miikkulainen et al., 2019; Real et al., 2017; 2020; Stanley & Miikkulainen, 2002; Zoph et al., 2018) can facilitate DNN design suited for nuisance-invariant inference. Nevertheless, without proper reasoning, most of the search space for link connectivity will be pointless.

In this paper, we propose a systematic automation framework called AutoBayes, which searches for the best inference graph model associated with a Bayesian graph model (also a.k.a. Bayesian network) well-suited to reproduce the training datasets. The proposed method automatically formulates various different Bayesian graphs by factorizing the joint probability distribution in terms of data, class label, subject identification (ID), and inherent latent representations. Given Bayesian graphs, some meaningful inference graphs are generated through the Bayes-Ball algorithm (Shachter, 2013) for pruning redundant links to achieve high-accuracy estimation. In order to promote robustness against nuisance variations such as inter-subject/session factors, the explored Bayesian graphs can provide

---

[1]For example of speech recognition, nuisance factors such as speaker's attributes and recording environment may change the task accuracy. For image recognition, ambient light conditions and image sensor conditions may become inherent nuisance factors. In the context of this paper, nuisance variations mainly refer to subject identities and biological states during recording sessions for physiological data learning.

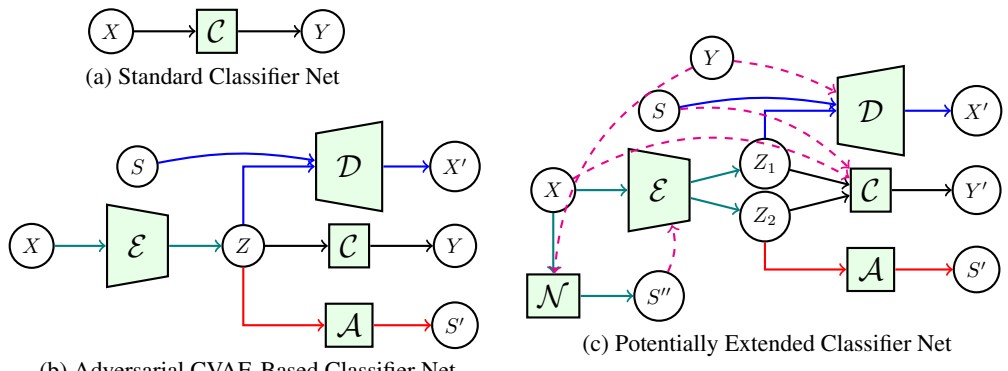

Figure 1: Inference methods to classify $Y$ given data $X$ under latent $Z$ and semi-labeled nuisance $S$.

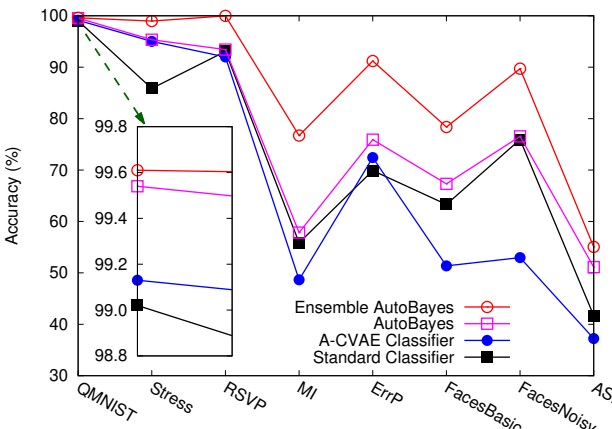

Figure 2: Model accuracy across different datasets. AutoBayes offers significant gain.

reasoning to use adversarial training with/without variational modeling and latent disentanglement. We demonstrate that AutoBayes can achieve excellent performance across various public datasets, in particular with an ensemble stacking of multiple explored graphical models.

## 2  KEY CONTRIBUTIONS

At the core of our methodology is the consideration of graphical models that capture the probabilistic relationship between random variables representing the data features $X$, task labels $Y$, nuisance variation labels $S$, and (potential) latent representations $Z$. The ultimate goal is to infer the task label $Y$ from the measured data feature $X$, which is hindered by the presence of nuisance variations (e.g., inter-subject/session variations) that are (partially) labelled by $S$. One may use a standard DNN to classify $Y$ given $X$ as shown in Fig. 1(a), without explicitly involving $S$ or $Z$. Although A-CVAE in Fig. 1(b) may offer nuisance-robust performance through adversarial disentanglement of $S$ from latent $Z$, there is no guarantee that such a model can perform well across different datasets. It is exemplified in Fig. 2 where A-CVAE outperforms the standard DNN model for some datasets (QMNIST, Stress, ErrP) while it does not for the other cases. This may be due to the underlying probabilistic relationship of the data varying across datasets. Our proposed framework can construct justifiable models, achieving higher performance for every dataset, as demonstrated in Fig. 2. It is verified that significant gain is attainable with ensemble methods of different Bayesian graphs which are explored in our AutoBayes. For example, our method with a relatively shallow architecture achieves 99.61% accuracy which is close to state-of-the-art performance in QMNIST dataset.

The main contributions of this paper over the existing works are five-fold as follows:

---

**Algorithm 1** Pseudocode for AutoBayes Framework

---

**Require:** Nodes set $\mathcal{V} = [Y, X, S_1, S_2, \ldots, S_n, Z_1, Z_2, \ldots, Z_m]$, where $Y$ denotes task labels, $X$ is a measurement data, $\mathcal{S} = [S_1, S_2, \ldots, S_n]$ are (potentially multiple) semi-supervised nuisance variations, and $\mathcal{Z} = [Z_1, Z_2, \ldots, Z_m]$ are (potentially multiple) latent vectors
**Ensure:** Semi-supervised training/validation datasets
1: **for all** permutations of node factorization from $Y$ to $X$ **do**
2:     Let $\mathcal{B}_0$ be the corresponding Bayesian graph for the permuted full-chain factorization
3:     **for all** combinations of link pruning on the full-chain Bayesian graph $\mathcal{B}_0$ **do**
4:         Let $\mathcal{B}$ be the corresponding pruned Bayesian graph
5:         Apply the Bayes-Ball algorithm on $\mathcal{B}$ to build a conditional independency list $\mathcal{I}$
6:         **for all** permutations of node factorization from $X$ to $Y$ **do**
7:             Let $\mathcal{F}_0$ be the factor graph corresponding to a full-chain conditional probability
8:             Prune all redundant links in $\mathcal{F}_0$ based on conditional independency $\mathcal{I}$
9:             Let $\mathcal{F}$ be the pruned factor graph
10:            Merge the pruned Bayesian graph $\mathcal{B}$ into the pruned factor graph $\mathcal{F}$
11:            Attach an adversary network $\mathcal{A}$ to latent nodes $\mathcal{Z}$ for $Z_k \perp \mathcal{S} \in \mathcal{I}$
12:            Assign an encoder network $\mathcal{E}$ for $p(\mathcal{Z}|\cdots)$ in the merged factor graph
13:            Assign a decoder network $\mathcal{D}$ for $p(x|\cdots)$ in the merged factor graph
14:            Assign a nuisance indicator network $\mathcal{N}$ for $p(\mathcal{S}|\cdots)$ in the merged factor graph
15:            Assign a classifier network $\mathcal{C}$ for $p(y|\cdots)$ in the merged factor graph
16:            Adversary train the whole DNN structure to minimize a loss function in (5)
17:        **end for**                              ▷ At most $(|\mathcal{V}| - 2)!$ combinations
18:     **end for**                                 ▷ At most $2^{|\mathcal{V}|(|\mathcal{V}|-1)/2}$ combinations
19: **end for**                                     ▷ At most $(|\mathcal{V}| - 2)!$ combinations
20: **return** the best model having highest task accuracy in validation sets

---

- AutoBayes automatically explores potential graphical models inherent to the data by combinatorial pruning of dependency assumptions (edges) and then applies Bayes-Ball to examine various inference strategies, rather than blindly exploring hyperparameters of DNN blocks.

- AutoBayes offers a solid reason of how to connect multiple DNN blocks to impose conditioning and adversary censoring for the task classifier, feature encoder, decoder, nuisance indicator and adversary networks, based on an explored Bayesian graph.

- The framework is also extensible to multiple latent representations and nuisances factors.

- Besides fully-supervised training, AutoBayes can automatically build some relevant graphical models suited for semi-supervised learning.

- Multiple graphical models explored in AutoBayes can be efficiently exploited to improve performance by ensemble stacking.

We note that this paper relates to some existing literature in AutoML, variational Bayesian inference (Kingma & Welling, 2013; Sohn et al., 2015; Louizos et al., 2015), adversarial training (Goodfellow et al., 2014; Dumoulin et al., 2016; Donahue et al., 2016; Makhzani et al., 2015; Lample et al., 2017; Creswell et al., 2017), and Bayesian network (Nie et al., 2018; Njah et al., 2019; Rohekar et al., 2018) as addressed in Appendix A.1 in more detail. Nonetheless, AutoBayes is a novel framework that diverges from AutoML, which mostly employs architecture tuning at a micro level. Our work focuses on exploring neural architectures at a macro level, which is not an arbitrary diversion, but a necessary interlude. Our method focuses on the relationships between the connections in a neural network's architecture and the characteristics of the data (Minsky & Papert, 2017). In addition to the macro-level structure learning of Bayesian network, our approach provides a new perspective in how to involve the adversarial blocks and to exploit multiple models for ensemble stacking.

## 3   AUTOBAYES

**AutoBayes Algorithm:**   The overall procedure of the AutoBayes algorithm is described in the pseudocode of Algorithm 1. The AutoBayes automatically constructs non-redundant inference factor graphs given a hypothetical Bayesian graph assumption, through the use of the Bayes-Ball algorithm.

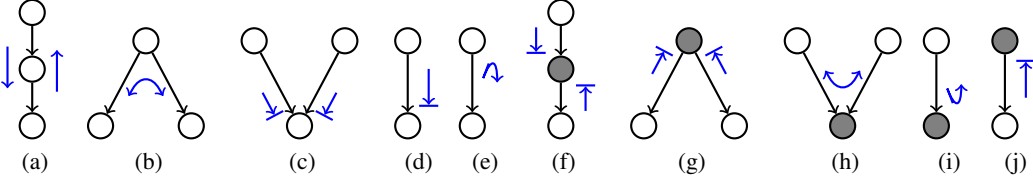

Figure 3: Bayes-Ball algorithm basic rules (Shachter, 2013). Conditional nodes are shaded.

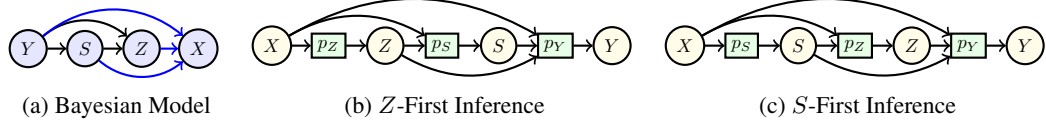

(a) Bayesian Model  (b) $Z$-First Inference  (c) $S$-First Inference

Figure 4: Full-chain Bayesian graph and inference models for $Z$-first or $S$-first factorizations.

Depending on the derived conditional independency and pruned factor graphs, DNN blocks for encoder $\mathcal{E}$, decoder $\mathcal{D}$, classifier $\mathcal{C}$, nuisance estimator $\mathcal{N}$ and adversary $\mathcal{A}$ are reasonably connected. The entire network is trained with variational Bayesian inference and adversarial training.

The Bayes-Ball algorithm (Shachter, 2013) facilitates an automatic pruning of redundant links in inference factor graphs through the analysis of conditional independency. Fig. 3 shows ten Bayes-Ball rules to identify conditional independency. Given a Bayesian graph, we can determine whether two disjoint sets of nodes are independent conditionally on other nodes through a graph separation criterion. Specifically, an undirected path is activated if a Bayes ball can travel along without encountering a stop symbol: $\longrightarrow\!|$ in Fig. 3. If there are no active paths between two nodes when some conditioning nodes are shaded, then those random variables are conditionally independent.

**Graphical Models:** We here focus on $4$-node graphs. Let $p(y, s, z, x)$ denote the joint probability distribution underlying the datasets for the four random variables, i.e., $Y$, $S$, $Z$, and $X$. The chain rule can yield the following factorization for a generative model from $Y$ to $X$ (note that at most $4!$ factorization orders exist including useless ones such as with reversed direction from $X$ to $Y$):

$$p(y, s, z, x) = p(y)p(s|y)p(z|s, y)p(x|z, s, y), \tag{1}$$

which is visualized in Bayesian graph of Fig. 4(a). The probability conditioned on $X$ can then be factorized, e.g., as follows (among $3!$ different orders of inference factorization for four-node graphs):

$$p(y, s, z|x) = \begin{cases} p(z|x)p(s|z, x)p(y|s, z, x), & \text{Z-first-inference} \\ p(s|x)p(z|s, x)p(y|z, s, x), & \text{S-first-inference} \end{cases} \tag{2}$$

which are marginalized to obtain the likelihood: $p(y|x) = \mathbb{E}_{s,z|x}\big[p(y, s, z|x)\big]$. The above two scheduling strategies in (2) are illustrated in factor graph models as in Figs. 4(b) and (c), respectively.

The graphical models in Fig. 4 do not impose any assumption of potentially inherent independency in datasets and hence are most generic. However, depending on the underlying independency in datasets, we may be able to prune some edges in those graphs. For example, if the data only follows the simple Markov chain of $Y - X$, while being independent of $S$ and $Z$, as shown in Fig. 5(a), all links except one between $X$ and $Y$ will be unreasonable in inference graphs of Figs. 4(b) and (c), that justifies the standard classifier model in Fig. 1(a). This implies that more complicated inference models such as A-CVAE can be unnecessarily redundant depending on the dataset. This motivates us to consider an extended AutoML framework which automatically explores the best pair of inference factor graph and corresponding Bayesian graph models matching dataset statistics besides the micro-scale hyperparameter tuning.

**Methodology:** AutoBayes begins with exploring any potential Bayesian graphs by cutting links of the full-chain graph in Fig. 4(a), imposing possible (conditional) independence. We then adopt the Bayes-Ball algorithm on each hypothetical Bayesian graph to examine conditional independence over

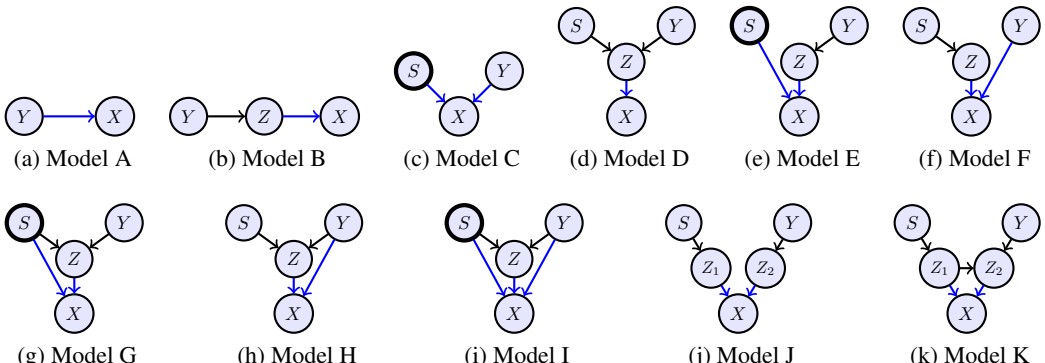

Figure 5: Example Bayesian graphs for data generative models under automatic exploration. Blue arrows indicate generative graph for decoder networks. Thick circled $S$ specifies the requirement of $S$-conditional decoder, which is less-convenient when learning unlabeled nuisance datasets.

different inference strategies, e.g., full-chain $Z$-/$S$-first inference graphs in Figs. 4(b)/(c). Applying Bayes-Ball justifies the reasonable pruning of the links in the full-chain inference graphs, and also the potential adversarial censoring when $Z$ is independent of $S$. This process automatically constructs the connectivity of inference, generative, and adversary blocks with sound reasoning.

Consider an example case when the data adheres to the following factorization:

$$p(y, s, z, x) = p(y)p(s|\cancel{y})p(z|\cancel{s}, y)\textcolor{blue}{p(x|z, s, \cancel{y})}, \tag{3}$$

where we explicitly indicate conditional independence by slash-cancellation from the full-chain case in (1). This corresponds to a Bayesian graphical model illustrated in Fig. 5(e). Applying the Bayes-Ball algorithm to the Bayesian graph yields the following conditional probability:

$$p(y, s, z|x) = p(z|x)p(s|z, x)p(y|z, \cancel{s}, \cancel{x}), \tag{4}$$

for the $Z$-first inference strategy in (2). The corresponding factor graph is then given in Fig. 6(c). Note that the Bayes-Ball also reveals that there is no marginal dependency between $Z$ and $S$, which provides the reason to use adversarial censoring to suppress nuisance information $S$ in the latent space $Z$. In consequence, by combining the Bayesian graph and factor graph, we automatically obtain A-CVAE model in Fig. 1(b). AutoBayes justifies A-CVAE under the assumption that the data follows the Bayesian model E in Fig. 5(e). As the true generative model is unknown, AutoBayes explores different Bayesian graphs like in Fig. 5 to search for the most relevant model. Our framework is readily applicable to graphs with more than 4 nodes to represent multiple $Y$, $S$, and $Z$. Models J and K in Fig. 5 are such examples having multiple latent factors $Z_1$ and $Z_2$. Despite the search space for AutoBayes will rapidly grow with the number of nodes, most realistic datasets do not require a large number of neural network blocks for macro-level optimization. See Appendix A.2 for more detailed descriptions for some Bayesian graph models to construct factor graphs like in Fig. 6. Also see discussions of graphical models suitable for semi-supervised learning in Appendix A.4.

**Training:** Given a pair of generative graph and inference graph, the corresponding DNN structures will be trained. For example of the generative graph model K in Fig. 5(k), one relevant inference graph Kz in Fig. 6(k) will result in the overall network structure as shown in Fig. 7, where adversary network is attached as $Z_2$ is (conditionally) independent of $S$. This 5-node graph model justifies a recent work on partially disentangled A-CVAE by Han et al. (2020). Each factor block is realized by a DNN, e.g., parameterized by $\theta$ for $p_\theta(z_1, z_2|x)$, and all of the networks except for adversarial network are optimized to minimize corresponding loss functions including $\mathcal{L}(\hat{y}, y)$ as follows:

$$\min_{\theta, \psi, \mu} \max_{\eta} \mathbb{E}\big[\mathcal{L}(\hat{y}, y) + \lambda_s \mathcal{L}(\hat{s}, s) + \lambda_x \mathcal{L}(\hat{x}', x) + \lambda_z \mathbb{KL}(z_1, z_2 \| \mathcal{N}(\mathbf{0}, \mathbf{I})) - \lambda_a \mathcal{L}(\hat{s}', s)\big], \tag{5}$$

$$(z_1, z_2) = p_\theta(x), \quad \hat{y} = p_\phi(z_1, z_2), \quad \hat{s} = p_\psi(z_1), \quad \hat{x}' = p_\mu(z_1), \quad \hat{s}' = p_\eta(z_2), \tag{6}$$

where $\lambda_*$ denotes a regularization coefficient, $\mathbb{KL}$ is the Kullback–Leibler divergence, and the adversary network $p_\eta(s'|z_2)$ is trained to minimize $\mathcal{L}(\hat{s}', s)$ in an alternating fashion (see the Adversarial Regularization paragraph below).

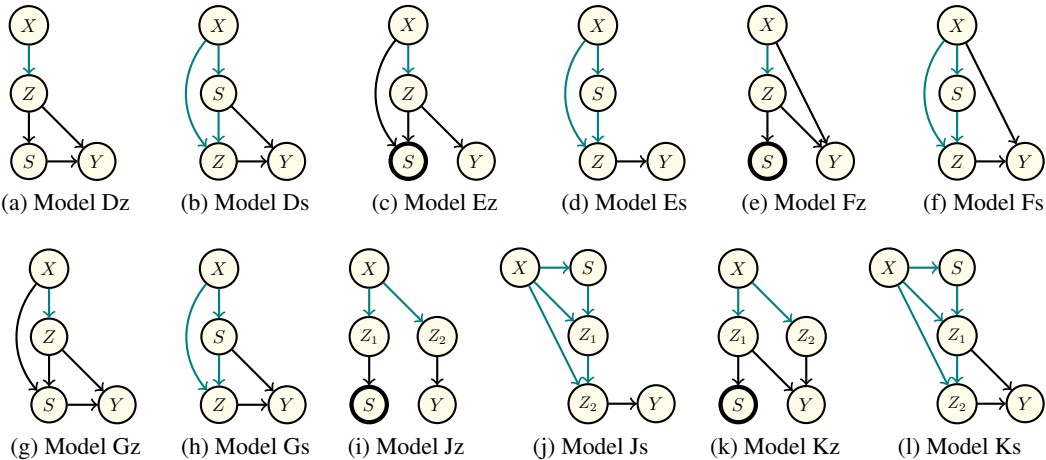

Figure 6: $Z$-first and $S$-first inference graph models relevant for generative models D–G, J, and K. Green arrows indicate feature extraction graph for encoder networks. Thick circled $S$ specifies the end node of inference, which is convenient when learning unlabeled nuisance datasets.

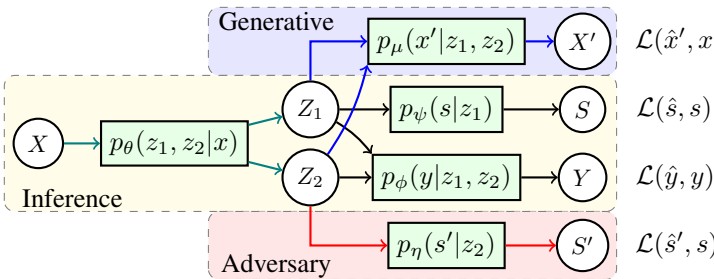

Figure 7: Overall network structure for pairing generative model K and inference model Kz.

The training objective can be formally understood from a likelihood maximization perspective, in manner that can be seen as a generalization of the VAE Evidence Lower Bound (ELBO) concept (Kingma & Welling, 2013). Specifically, it can be viewed as the maximization of a variational lower bound of the likelihood $p_\Phi(x, y, s)$ that is implicitly defined and parameterized by the networks, where $\Phi$ represents the collective parameters of the network modules (e.g., $\Phi = (\phi, \psi, \mu)$ in the example of equation 5) that specify the generative model $p_\Phi(x, y, s|z)$, which implies the likelihood $p_\Phi(x, y, s)$, as given by

$$ p_\Phi(x, y, s) = \int p_\Phi(x, y, s|z)p(z)\, dz. $$

However, since this expression is generally intractable, we introduce $q_\theta(z|x, y, s)$ as a variational approximation of the posterior $p_\Phi(z|x, y, s)$ implied by the generative model (Kingma & Welling, 2013; Ranganath et al., 2014):

$$ \frac{1}{n}\sum_{i=1}^{n} \log p_\Phi(x_i, y_i, s_i) = \frac{1}{n}\sum_{i=1}^{n}\left[\log p_\Phi(x_i, y_i, s_i|z_i) - \log \frac{q_\theta(z_i|x_i, y_i, s_i)}{p(z_i)} + \log \frac{q_\theta(z_i|x_i, y_i, s_i)}{p_\Phi(z_i|x_i, y_i, s_i)}\right] $$

$$ \approx \frac{1}{n}\sum_{i=1}^{n}\left[\log p_\Phi(x_i, y_i, s_i|z_i)\right] - \mathbb{KL}(q_\theta(z|x, y, s)\|p(z)) + \mathbb{KL}(q_\theta(z|x, y, s)\|p_\Phi(z|x, y, s)) $$

$$ \geq \frac{1}{n}\sum_{i=1}^{n}\left[\log p_\Phi(x_i, y_i, s_i|z_i)\right] - \mathbb{KL}(q_\theta(z|x, y, s)\|p(z)), \tag{7} $$

where the samples $z_i \sim q_\theta(z|x_i, y_i, s_i)$ are drawn for each training tuple $(x_i, y_i, s_i)$, and the final inequality follows from the non-negativity of KL divergence.

Table 1: Parameters of Public Dataset Under Investigation

| Dataset | Modality | Dimension | Nuisance ($|S|$) | Classes ($|Y|$) | Samples | Reference |
|---------|----------|-----------|------------------|-----------------|---------|-----------|
| QMNIST | Image | $28 \times 28$ | 836 | 10 | 70,000 | Yadav & Bottou (2019) |
| Stress | Temperature etc. | $7 \times 1$ | 20 | 4 | 24,000 | Birjandtalab et al. (2016) |
| RSVP | EEG | $16 \times 128$ | 10 | 4 | 41,400 | Orhan et al. (2012) |
| MI | EEG | $64 \times 480$ | 106 | 4 | 9,540 | Goldberger et al. (2000) |
| ErrP | EEG | $56 \times 250$ | 27 | 2 | 9,180 | Margaux et al. (2012) |
| Faces Basic | ECoG | $31 \times 400$ | 14 | 2 | 4,100 | Miller et al. (2015; 2016) |
| Faces Noisy | ECoG | $39 \times 400$ | 7 | 2 | 2,100 | Miller et al. (2015; 2017) |
| ASL | EMG | $16 \times 100$ | 5 | 33 | 9,900 | Günay et al. (2019) |

Ultimately, the minimization of our training loss function corresponds to the maximization of the lower bound in (7), which corresponds to maximizing the likelihood of our implicit generative model, while also optimizing the variational posterior $q_\theta(z|x, y, s)$ toward the actual posterior for the latent representation $p_\Phi(z|x, y, s)$, since the gap in the bound is given by $\mathbb{KL}(q_\theta(z|x, y, s) \| p_\Phi(z|x, y, s))$. Further factoring of $\log p_\Phi(x, y, s|z)$ yields the multiple loss-terms and network modules.

**Adversarial Regularization:** We can utilize adversarial censoring when $Z$ and $S$ should be marginally independent, e.g., such as in Fig. 1(b) and Fig. 7, in order to reinforce the learning of a representation $Z$ that is disentangled from the nuisance variations $S$. This is accomplished by introducing an adversarial network that aims to maximize a parameterized approximation $q(s|z)$ of the likelihood $p(s|z)$, while this likelihood is also incorporated into the loss for the other modules with a negative weight. The adversarial network, by maximizing the log likelihood $\log q(s|z)$, essentially maximizes a lower-bound of the mutual information $\mathbb{I}(S; Z)$, and hence the main network is regularized with the additional term that corresponds to minimizing this estimate of mutual information. This follows since the log-likelihood maximized by the adversarial network is given by

$$\mathbb{E}[\log q(s|z)] = \mathbb{I}(S; Z) - \mathbb{H}(S) - \mathbb{KL}\big(p(s|z) \| q(s|z)\big), \tag{8}$$

where the entropy $\mathbb{H}(S)$ is constant.

**Ensemble Learning:** We further introduce ensemble methods to make best use of all Bayesian graph models explored by the AutoBayes framework without wasting lower-performance models. Ensemble stacked generalization works by stacking the predictions of the base learners in a higher level learning space, where a meta learner corrects the predictions of base learners (Wolpert, 1992). Subsequent to training base learners, we assemble the posterior probability vectors of all base learners together to improve the prediction. We compare the predictive performance of a logistic regression (LR) and a shallow multi-layer perceptron (MLP) as an ensemble meta learner to aggregate all inference models. See Appendix A.5 for more detailed description of the stacked generalization.

## 4 EXPERIMENTAL EVALUATION

**Datasets:** We experimentally demonstrate the performance of AutoBayes for publicly available datasets as listed in Table 1. Note that they cover a wide variety of data size, dimensionality, subject scale, and class levels as well as sensor modalities including image, EEG, EMG, and electrocorticography (ECoG). See more detailed information of each dataset in Appendix A.6.

**Model Implementation:** All models were trained with a minibatch size of 32 and using the Adam optimizer with an initial learning rate of 0.001. The learning rate is halved whenever the validation loss plateaus. A compact convolutional neural network (CNN) with 4 layers is employed as an encoder network $\mathcal{E}$ to extract features from $C \times T$ data. Each convolution is followed by batch normalization (BN) and rectified linear unit (ReLU) activation. The AutoBayes chooses either a deterministic latent encoder or variational latent encoder under Gaussian prior. The original data is reconstructed by a decoder network $\mathcal{D}$ that applies transposed convolutions. All of our experiments were run for 20 epochs on NVIDIA Tesla K80 12GB GPU. See Appendix A.7 for more details.

**Results:** Fig. 8(a) shows the performance of QMNIST across 39 different inference models explored in AutoBayes including 2 ensemble models. Over 37 base models, some outperforms the standard

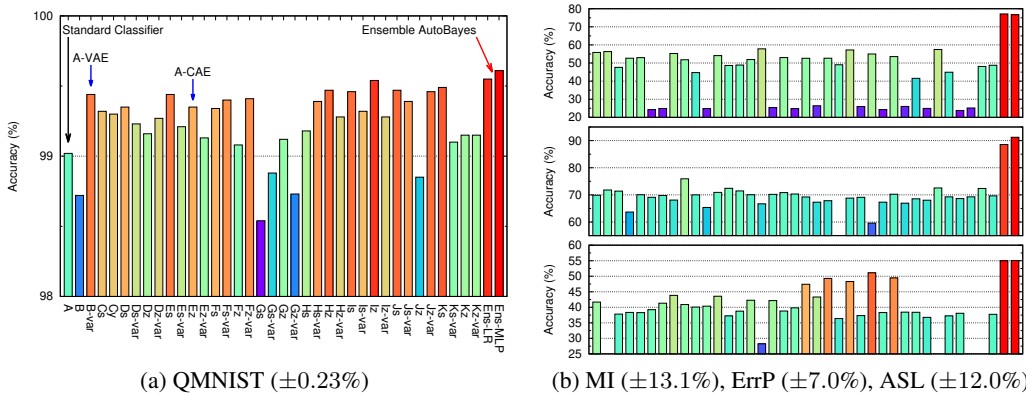

(a) QMNIST (±0.23%)

(b) MI (±13.1%), ErrP (±7.0%), ASL (±12.0%)

Figure 8: Task classification accuracy across different graphical models (with standard deviation).

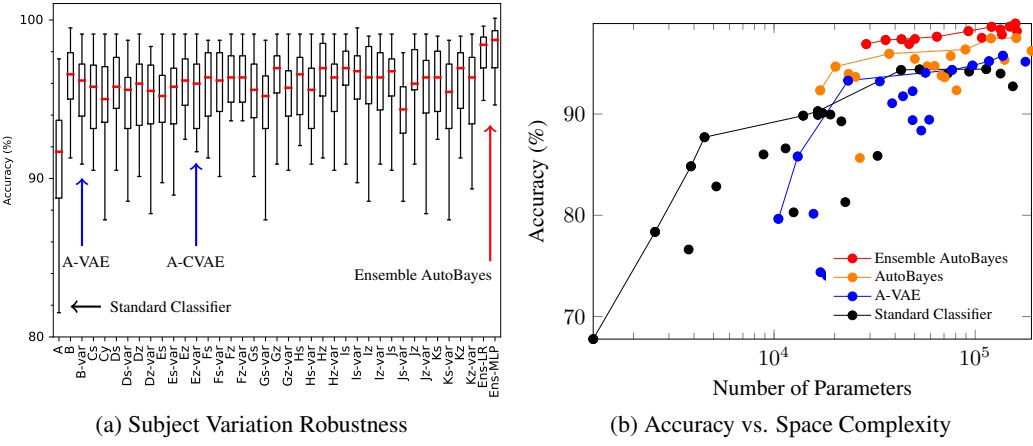

(a) Subject Variation Robustness

(b) Accuracy vs. Space Complexity

Figure 9: Task classification accuracy for Stress dataset.

classifier model A, whereas the rest of the models underperform. We observe a large gap of 1.0% between the best and worst models with a standard deviation of 0.23% across all Bayesian graph models. This indicates that we may have a potential risk that one particular model may lose up to 1.0% accuracy if we do not explore different models.

Similar behaviors with a huge deviation can be seen for different datasets as shown in Fig. 8(b). It was shown that the best inference strategy highly depends on datasets. Specifically, the best model at one dataset does not perform best for different datasets. This suggests that we must consider different inference strategies for each target dataset and our AutoBayes provides such an adaptive framework across datasets. More detailed results are found in Appendix A.8.

Remarkably, the ensemble of base learners further enhances the performance regardless of the choice from LR or MLP as the meta learner, as illustrated in Fig. 2 across all the datasets. For some low-performing datasets such as ErrP, MI and Faces (Noisy), ensemble learning significantly improves the accuracy by 15.3%, 19.3% and 13.2% at the expense of more storage and computational resources.

Exploring different models has actually a significant benefit in improving nuisance robustness as shown in Fig. 9(a), where box-whisker plots are present to show the quartile distribution of the subject variation for the Stress dataset having $|S| = 20$ users. We can observe that the standard classification (Model A) has a wider distribution; the best subject achieves an accuracy grater than 96%, whereas the worst-case user has lower than 82% accuracy. Except for model A, the other models from B to Kz take the subject ID ($S$) into consideration to extract nuisance-robust feature, which leads to significant improvement for the worst-case user performance not only for the mean or median. The

Table 2: Task classification performance of AutoBayes compared to state-of-the-art.

| Method | QMNIST | Stress | RSVP | MI | ErrP | Faces Basic | Faces Noisy | ASL |
|---|---|---|---|---|---|---|---|---|
| Ensemble AutoBayes (Meta-MLP) | 99.61 | **98.98** | **99.99** | 76.71 | **91.21** | **78.36** | **89.71** | **55.06** |
| Ensemble AutoBayes (Meta-LR) | 99.55 | 98.96 | 99.98 | **77.14** | 88.54 | 75.68 | 88.40 | 54.98 |
| Best of AutoBayes | 99.54 | 95.35 | 93.42 | 57.83 | 75.91 | 67.31 | 76.58 | 51.12 |
| State-of-the-art (SOTA) | **99.84** | 85.30 | 71.60 | 63.8 | 48.80 | — | — | — |

ensemble stacking further improves the subject variation robustness, achieving the worst-case user performance of at least $94\%$. Additional results per user are found in Appendix A.9.

Despite the performance gain, the nuisance-robust models tend to have higher complexity. Fig. 9(b) shows the trade-off between the accuracy and the space complexity. Here, we varied the number of hidden layers and hidden nodes for the models A, B, and Js to adjust the space complexity. The Pareto front over the finite set of DNN configurations is indicated with lines. It is observed that the standard classifier model A has superior performance only at low complexity regimes, while it does not improve performance beyond $95\%$ accuracy even with increased complexity. The Pareto front of AutoBayes is thus better than the individual models at higher accuracy regimes. See Appendix A.10 for an additional analysis of time complexity.

We finally compare the performance of AutoBayes with the benchmark competitor models from (Byerly et al., 2020; Han et al., 2020; Özdenizci et al., 2019c;b;a) in Table 2. It can be seen that AutoBayes outperforms the state-of-the-art in all datasets except QMNIST. Consequently, we can see a great advantage of AutoBayes with exploring different graphical models. Even for QMNIST, AutoBayes meta-MLP model, achieving $99.61\%$ accuracy, ranks 17 in the published leaderboard. Note that performing better than $99.84\%$ is nearly impossible, since some numbers are illegible or mislabeled. Also note that we have not specifically designed AutoBayes architecture for image classification but for spatio-temporal signal applications and hyper-parameters were not fully optimized yet.

AutoBayes can be readily integrated with AutoML to optimize any hyperparameters of individual DNN blocks. Nevertheless, as our primary objective was to show a proof-of-concept benefit from solely graphical model exploration of AutoBayes, we leave more rigorous analysis to optimize DNN parameters such as network depths, widths, activation, augmentation, etc. as a future work.

## 5 CONCLUSION AND FUTURE WORK

We proposed a new concept called AutoBayes which explores various different Bayesian graph models to facilitate searching for the best inference strategy, suited for nuisance-robust deep learning. With the Bayes-Ball algorithm, our method can automatically construct reasonable link connections among classifier, encoder, decoder, nuisance estimator and adversary DNN blocks. As a proof-of-concept analysis, we demonstrated the benefit of AutoBayes for various public datasets. We observed a huge performance gap between the best and worst graph models, implying that the use of one particular model without graph exploration can potentially suffer a poor classification result. In addition, the best model for one dataset does not always perform best for different data, which encourages us to use AutoBayes for adaptive model generation given target datasets. We further improved the performance approaching the state-of-the-art accuracy by exploiting multiple graphical models explored in AutoBayes through the use of ensemble stacking. The ensemble AutoBayes offers significant gain in nuisance robustness by improving the worst-case user performance. Even though additional computations are required, we showed that AutoBayes can still achieve the superior Pareto front in the trade-off between complexity and accuracy. We are extending the AutoBayes framework to integrate AutoML to optimize hyperparameters of each DNN block. How to handle the exponentially growing search space of possible Bayesian graphs along with the number of random variables remains a challenging future work. It should require more sophisticated metrics like Bayesian information criterion for efficient graph exploration.

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

## APPENDICES

### A.1 RELATED WORK

We note that this paper relates to some existing literature as follows.

- **AutoML:** Searching DNN models with hyperparameter optimization has been intensively investigated in a framework called AutoML (Ashok et al., 2017; Brock et al., 2017; Cai et al., 2017; He et al., 2018; Miikkulainen et al., 2019; Real et al., 2017; 2020; Stanley & Miikkulainen, 2002; Zoph et al., 2018). The automated methods include architecture search (Zoph et al., 2018; Real et al., 2017; He et al., 2018; Real et al., 2020), learning rule design (Bayer et al., 2009; Jozefowicz et al., 2015), and augmentation exploration (Cubuk et al., 2019; Park et al., 2019). Most work used either evolutionary optimization or reinforcement learning framework to adjust hyperparameters or to construct network architecture from pre-selected building blocks. (Miconi, 2016) gradually increases the size of an RNN starting from only one node by incorporating structural parameters into model training, which are optimized along with the model weights. (Zoph & Le, 2016) uses reinforcement learning to find the optimal neural network architecture based on actor-critic framework. The method uses an LSTM as a controller and critic to explore the hyperparameter configurations for each layer (number of filters, kernel size and stride) based on the validation error of the output architecture that corresponds to reward. The recent AutoML-Zero (Real et al., 2020) considers an extension to preclude human knowledge and insights for fully automated designs from scratch.

- **Variational Bayesian Inference:** The VAE (Kingma & Welling, 2013) introduced variational Bayesian inference methods, incorporating autoassociative architectures, where generative and inference models can be learned jointly. This method was extended with the CVAE (Sohn et al., 2015), which introduces a conditioning variable that could be used to represent nuisance variations, and a regularized VAE in (Louizos et al., 2015), which considers disentangling the nuisance variable from the latent representation.

- **Adversarial Training:** The concept of adversarial networks was introduced with Generative Adversarial Networks (GAN) (Goodfellow et al., 2014), and has been adopted into myriad applications. The simultaneously discovered Adversarially Learned Inference (ALI) (Dumoulin et al., 2016) and Bidirectional GAN (BiGAN) (Donahue et al., 2016) propose an adversarial approach toward training an autoencoder. Adversarial training has also been combined with VAE to regularize and/or disentangle the latent representations (Makhzani et al., 2015; Lample et al., 2017; Creswell et al., 2017).

- **Bayesian Network Structure Learning:** Deep Bayesian network (Nie et al., 2018; Njah et al., 2019; Rohekar et al., 2018) has been studied to learn probabilistic relationships between random variables. Learning model structure of a Bayesian network is a problem that has long been studied, e.g., recovery algorithm (Rebane & Pearl, 2013), scoring methods (Campos, 2006), and constraint methods (Scutari, 2014; Pearl et al., 2000). Scoring methods commonly use the posterior probability of the Bayesian network given training data, such as Bayesian information criterion (BIC). Although the complexity of an exhaustive search is superexponential in the number of variables, recent approaches (Cussens et al., 2017) showed capability to learn structure of Bayesian network with up to 100 variables using integer programming. Constraint-based methods use conditional independence tests between pairs of variables, commonly mutual information test or the Student's t-test for correlation. All constraint-based methods entail three phases: i.e., (i) learning Markov blankets of each variable, (ii) learning neighbors (parents and children) of each variable that identifies which arcs are present in a Bayesian network, and (iii) establishing arc directions.

Compared to the existing AutoML literature, our method provides more systematic framework to explore justifiable network architectures from a macro view. Although related Bayesian network was studied to design DNN architecture, our method extends it to realize nuisance robustness by reasonably involving adversarial networks. In addition, ensemble stacking was first introduced in AutoML framework where multiple architectures can be reused to improve the performance over every individual model.

## A.2 BAYESIAN GRAPH AND INFERENCE MODELS

Given measurement data, we never know the true joint probability beforehand, and therefore we shall assume one of several possible generative models. AutoBayes aims to explore such potential graph models to match the measurement distributions. As the maximum possible number of graphical models is huge even for a four-node case involving $Y$, $S$, $Z$ and $X$, we restrict our focus to a few meaningful graphs-of-interest shown in Fig. 5. Each Bayesian graph corresponds to the following assumption of the joint probability factorization ($p(x|\cdots)$ term specifies a generative model of $X$):

$$p(y,s,z,x) = \begin{cases} p(y)p(s|\cancel{y})p(z|\cancel{s},\cancel{y})p(x|\cancel{z},\cancel{s},y), & \text{Model-A} \\ p(y)p(s|\cancel{y})p(z|\cancel{s},y)p(x|z,\cancel{s},\cancel{y}), & \text{Model-B} \\ p(y)p(s|\cancel{y})p(z|\cancel{s},\cancel{y})p(x|\cancel{z},s,y), & \text{Model-C} \\ p(y)p(s|\cancel{y})p(z|s,y)p(x|z,\cancel{s},\cancel{y}), & \text{Model-D} \\ p(y)p(s|\cancel{y})p(z|\cancel{s},y)p(x|z,s,\cancel{y}), & \text{Model-E} \\ p(y)p(s|\cancel{y})p(z|s,\cancel{y})p(x|z,\cancel{s},y), & \text{Model-F} \\ p(y)p(s|\cancel{y})p(z|s,y)p(x|z,s,\cancel{y}), & \text{Model-G} \\ p(y)p(s|\cancel{y})p(z|s,y)p(x|z,\cancel{s},y), & \text{Model-H} \\ p(y)p(s|\cancel{y})p(z|s,y)p(x|z,s,y), & \text{Model-I} \\ p(y)p(s|\cancel{y})p(z_1|s,\cancel{y})p(z_2|\cancel{z_1},\cancel{s},y)p(x|z_2,z_1,\cancel{s},\cancel{y}), & \text{Model-J} \\ p(y)p(s|\cancel{y})p(z_1|s,\cancel{y})p(z_2|z_1,\cancel{s},y)p(x|z_2,z_1,\cancel{s},\cancel{y}), & \text{Model-K} \end{cases} \quad (9)$$

where we explicitly indicate independence by slash-cancelled factors from the full-chain case in equation 1. Blue-colored terms correspond to the blue arrows in Figs. 5 for generative graph of decoder networks. Depending on the assumed Bayesian graph, the relevant inference strategy will vary as some variables may be conditionally independent, which enables pruning links in the inference factor graphs. As shown in Fig. 6, the reasonable inference graph model can be automatically generated by the Bayes-Ball algorithm (Shachter, 2013) on each Bayesian graph hypothesis inherent in datasets. Specifically, the conditional probability $p(y,s,z|x)$ can be obtained for each model as below.

**Bayesian Graph Model A (Direct Markov):** The simplest model between $X$ and $Y$ would be single Markov chain without any dependency of $S$ and $Z$, shown in Bayesian graph of Fig. 5(a). This model puts an assumption that the data are nuisance-invariant. For this case, there is no reason to employ complicated inference models such as A-CVAE since most factors will be independent as $p(y,s,z|x) = p(z|\cancel{x})p(s|\cancel{z},\cancel{x})p(y|\cancel{s},\cancel{z},x)$. We hence should use a standard classification method, as in Fig. 1(a), to infer $Y$ given $X$, based on the inference model $p(y|x)$ without involving $S$ and $Z$.

**Bayesian Graph Model B (Markov Latent):** Assuming a latent $Z$ can work in a Markov chain of $Y - Z - X$ shown in Fig. 5(b), we obtain a simple inference model: $p(y,s,z|x) = p(z|x)p(s|\cancel{z},\cancel{x})p(y|\cancel{s},z,\cancel{x})$. Note that this model assumes independence between $Z$ and $S$, and thus adversarial censoring (Makhzani et al., 2015; Creswell et al., 2017; Lample et al., 2017) can make it more robust against nuisance. This model is hence based on A-VAE.

**Bayesian Graph Model C (Subject-Dependent):** We may model the case when the data $X$ directly depends on subject $S$ and task $Y$, shown in Fig. 5(c). For this case, we may consider the corresponding inference models due to the Bayes-Ball:

$$p(y,s,z|x) = \begin{cases} p(s|x)p(z|\cancel{s},\cancel{x})p(y|s,\cancel{z},x), & \text{Model-Cs} \\ p(y|x)p(s|y,x)p(z|\cancel{s},\cancel{y},\cancel{x}). & \text{Model-Cy} \end{cases} \quad (10)$$

Note that this model does not depend on $Z$, and thus $Z$-first inference strategy reduces to $S$-first model. As a reference, we here consider additional $Y$-first inference strategy to evaluate the difference.

**Bayesian Graph Model D (Latent Summary):** Another graphical model is shown in Fig. 5(d), where a latent space bridges all other random variables. Bayes-Ball yields the following models:

$$p(y,s,z|x) = \begin{cases} p(z|x)p(s|z,\cancel{x})p(y|s,z,\cancel{x}), & \text{Model-Dz} \\ p(s|x)p(z|s,x)p(y|z,s,\cancel{x}), & \text{Model-Ds} \end{cases} \quad (11)$$

whose graphical models are depicted in Figs. 6(a) and (b), respectively.

**Bayesian Graph Model E (Task-Summary Latent):** Another graphical model involving latent variables is shown in Fig. 5(e), where a latent space only summarizes $Y$. Bayes-Ball yields the following inference models:

$$p(y, s, z|x) = \begin{cases} p(z|x)p(s|z,x)p(y|z,\cancel{s},\cancel{x}), & \text{Model-Ez} \\ p(s|x)p(z|s,x)p(y|\cancel{s},z,\cancel{x}), & \text{Model-Es} \end{cases} \qquad (12)$$

which are illustrated in Figs. 6(c) and (d). Note that the generative model E has no marginal dependency between $Z$ and $S$, which provides the reason to use adversarial censoring to suppress nuisance information $S$ in the latent space $Z$. In addition, because the generative model of $X$ is dependent on both $Z$ and $S$, it is justified to employ the A-CVAE classifier shown in Fig. 1(b).

**Bayesian Graph Model F (Subject-Summary Latent):** Consider Fig. 5(f), where a latent variable summarizes subject information $S$. The Bayes-Ball provides the inference graphs shown in Figs. 6(e) and (f), which respectively correspond to:

$$p(y, s, z|x) = \begin{cases} p(z|x)p(s|z,\cancel{x})p(y|\cancel{s},x,z), & \text{Model-Fz} \\ p(s|x)p(z|s,x)p(y|x,\cancel{s},z). & \text{Model-Fs} \end{cases} \qquad (13)$$

**Bayesian Graph Model G:** Letting the joint distribution follow the model G in Fig. 5(g), we obtain the following inference models via the Bayes-Ball:

$$p(y, s, z|x) = \begin{cases} p(z|x)p(s|z,x)p(y|s,z,\cancel{x}), & \text{Model-Gz} \\ p(s|x)p(z|s,x)p(y|z,s,\cancel{x}), & \text{Model-Gs} \end{cases} \qquad (14)$$

whose graphical models are described in Figs. 6(g) and (h). Note that the inference model Gs in Fig. 6(h) is identical to the inference model Ds in Fig. 6(b). Although the inference graphs Gs and Ds are identical, the generative model of $X$ is different as shown in Figs. 5(g) and (d). Specifically, VAE decoder for the model G should feed $S$ along with variational latent space $Z$, and thus using CVAE is justified for the model G but D. This difference of the generative models can potentially make a different impact on the performance of inference despite the inference graph alone is identical.

**Bayesian Graph Models H and I:** Both the generative models H and I shown in Figs. 5(h) and (i) have the fully-connected inference strategies as given in (2), whose graphs are shown in Figs. 4(b) and (c), respectively, since no useful conditional independency can be found with the Bayes-Ball. Analogous to the relation of models Ds and Gs, the inference graph can be identical for Bayesian graphs H and I, whereas the generative model of $X$ is different as shown in Figs. 5(h) and (i).

**Bayesian Graph Model J (Disentangled Latent):** We can also consider multiple latent vectors to generalize the Bayesian graph with more vertices. We here focus on two such examples of graph models with two-latent spaces as shown in Figs. 5(j) and (k). Those models are identical class of the model D, except that a single latent $Z$ is disentangled into two parts $Z_1$ and $Z_2$, respectively associated with $S$ and $Y$. Given the Bayesian graph of Fig. 5(j), the Bayes-Ball yields some inference strategies including the following two models:

$$p(y, s, z_1, z_2|x) = \begin{cases} p(z_1, z_2|x)p(s|z_1,\cancel{z_2},\cancel{x})p(y|\cancel{s},\cancel{z_1},z_2,\cancel{x}), & \text{Model-Jz} \\ p(s|x)p(z_1|s,x)p(z_2|\cancel{s},z_1,x)p(y|\cancel{s},\cancel{z_1},z_2,\cancel{x}), & \text{Model-Js} \end{cases} \qquad (15)$$

which are shown in Figs. 6(i) and (j). Note that $Z_2$ is marginally independent of the nuisance variable $S$, which encourages the use of adversarial training to be robust against subject/session variations.

**Bayesian Graph Model K (Conditionally Disentangled Latent):** Another modified model in Fig. 5(k) linking $Z_1$ and $Z_2$ yields the following inference models:

$$p(y, s, z_1, z_2|x) = \begin{cases} p(z_1, z_2|x)p(s|z_1,\cancel{z_2},\cancel{x})p(y|\cancel{s},z_1,z_2,\cancel{x}), & \text{Model-Kz} \\ p(s|x)p(z_1|s,x)p(z_2|\cancel{s},z_1,x)p(y|\cancel{s},z_1,z_2,\cancel{x}), & \text{Model-Ks} \end{cases} \qquad (16)$$

as shown in Figs. 6(k) and (l). The major difference from the model J lies in the fact that the inference graph should use $Z_1$ along with $Z_2$ to infer $Y$.

### A.3 BACKGROUND ON VARIATIONAL BAYESIAN INFERENCE

**Variational AE** AutoBayes may automatically construct autoencoder architecture when latent variables are involved, e.g., for the model E in Fig. 5(e). For this case, $Z$ represents a stochastic node to marginalize out for $X$ reconstruction and $Y$ inference, and hence VAE will be required. In contrast to vanilla autoencoders, VAE uses variational inference by assuming a marginal distribution for latent $p(z)$. In variational approach, we reparameterize $Z$ from a prior distribution such as the normal distirbution to marginalize. Depending on the Bayesian graph models, we can also consider reparametering semi-supervision on $S$ (i.e., incorporating a reconstruction loss for $S$) as a conditioning variable. Conditioning on $Y$ and/or $S$ should depend on consistency with the graphical model assumptions. Since VAE is a special case of CVAE, we will go into further detail about the more general CVAE below.

**Conditional VAE** When $X$ is directly dependent on $S$ or $Y$ along with $Z$ in the Bayesian graph, the AutoBayes gives rise the CVAE architecture, e.g., for the models E/F/G/H/I in Fig. 5. For those generative models, the decoder DNN needs to feed $S$ or $Y$ as a conditioning parameter. Even for other Bayesian graphs, the $S$-first inference strategy will require conditional encoder in CVAE, e.g., the models Ds/Es/Fs/Gs/Js/Ks in Fig. 6, where latent $Z$ depends on $S$.

Consider the case when $S$ plays as the conditioning variable in a data model with the factorization:

$$p(s, x, z) = p(s)p(z)p(x|s, z), \tag{17}$$

where we directly parameterize $p(x|s, z)$, set $p(z)$ to something simple (e.g., isotropic Gaussian), and leave $p(s)$ arbitrary (since it will not be directly used). The CVAE is trained according to maximizing the likelihood of data tuples $(s, x)$ with respect to $p(x|s)$, which is given by

$$p(x|s) = \int p(x|s, z)p(z)\,\mathrm{d}z, \tag{18}$$

which is intractable to compute exactly given the potential complexity of the parameterization of $p(x|s, z)$. While it could be possible to approximate the integration with sampling of $Z$, the crux of the VAE approach is to utilize a variational lower-bound of the likelihood that involves a variational approximation of the posterior $p(z|s, x)$ implied by the generative model. With $q(z|s, x)$ representing the variational approximation of the posterior, the Evidence Lower-Bound (ELBO) is given by

$$\log p(x|s) \geq \mathbb{E}_{z \sim q(z|s,x)}[\log p(x|s, z)] - \mathbb{KL}\big(q(z|s, x)\|p(z)\big). \tag{19}$$

The parameterization of the variational posterior $q(z|s, x)$ may also be decomposed into parameterized components, e.g., $q(z|s, x) = q(s|x)q(z|s, x)$ such as in the $S$-first models shown in Fig. 6. Such decomposition also enables the possibility of semi-supervised training, which can be convenient when some of the variables, such as the nuisances variations, are not always labeled. For data tuples that include $s$, the likelihood $q(s|x)$ can also be directly optimized, and the given value for $s$ is used an input to the computation of $q(z|s, x)$. However, for tuples where $s$ is missing, the component $q(s|x)$ can be used to generate an estimate of $s$ to be input to $q(z|s, x)$. We further discuss semi-supervised learning and the sampling methods for categorical nuisance variables in Appendix A.4 below.

### A.4 SEMI-SUPERVISED LEARNING: CATEGORICAL SAMPLING

**Graphical Models for Semi-Supervised Learning** Nuisance values $S$ such as subject ID or session ID may not be always available for typical physiological datasets, in particular for the testing phase of an HMI system deployment with new users, requiring semi-supervised methods. We note that some graphical models are well-suited for such semi-supervised training. For example, among the Bayesian graph models in Fig. 5, the models C/E/G/I require the nuisance $S$ to reproduce $X$. If no ground-truth labels of $S$ are available, we need to marginalize $S$ across all possible categories for the decoder DNN $\mathcal{D}$. Even for other Bayesian graphs, the corresponding inference factor graphs in Fig. 6 may not be convenient for the semi-supervised settings. Specifically, for models Ez/Fz/Jz/Kz have an inference of $S$ at the end node, whereas the other inference models use inferred $S$ for subsequent inference of other parameters. If $S$ is missing or unknown as a semi-supervised setting, those inference graphs having $S$ in a middle node are inconvenient as we need sampling over all possible nuisance categories. For instance, the model Kz shown in Fig. 7 does not need $S$ marginalization, and thus readily applicable to semi-supervised datasets.

**Variational Categorical Reparameterization**    In order to deal with the issue of categorical sampling, we can use the Gumbel-Softmax reparameterization trick (Jang et al., 2016), which enables differentiable approximation of one-hot encoding. Let $[\pi_1, \pi_2, \ldots, \pi_{|S|}]$ denote a target probability mass function for the categorical variable $S$. Let $g_1, g_2, \ldots, g_{|S|}$ be independent and identically distributed samples drawn from the Gumbel distribution $\mathrm{Gumbel}(0, 1)$.[2] Then, generate an $|S|$-dimensional vector $\hat{s} = [\hat{s}_1, \hat{s}_2, \ldots, \hat{s}_{|S|}]$ according to

$$\hat{s}_k = \frac{\exp((\log(\pi_k) + g_k)/\tau)}{\sum_{i=1}^{|S|} \exp((\log(\pi_i) + g_i)/\tau)}, \tag{20}$$

where $\tau > 0$ is a softmax temperature. As the softmax temperature $\tau$ approaches 0, samples from the Gumbel-Softmax distribution become one-hot and the distribution becomes identical to the target categorical distribution. The temperature $\tau$ is usually decreased across training epochs as an annealing technique, e.g., with exponential decaying.

## A.5    Ensemble Learning: Stacked Generalization

To achieve higher predictive performance, we construct ensembles from the output posterior class probabilities of all graphical models. Let $\mathcal{D}_0 = \{(x_n, y_n, s_n)|n = 1 : N\}$ denote a data set, where $x_n$ is a data instance, $y_n$ is the task label, $s_n$ is the nuisance (subject) label and $N$ is the number of samples in the dataset. We randomly split the data into training set $\mathcal{D}_{\mathrm{train}}$ and validation set $\mathcal{D}_{\mathrm{test}}$. Given 37 graphical models, which we call base learners, we induce a decision algorithm $\mathcal{M}_k$, for $k = 1, \ldots, 37$ by invoking the $k$th graphical model on the data in $\mathcal{D}_{\mathrm{train}}$. For each $x_n$ in $\mathcal{D}_{\mathrm{train}}$, graphical model $\mathcal{M}_k$ generates a class probability vector for task and nuisance label prediction. Let $P_{ky}(x_n) = \{P(y_1|x_n), \ldots, P(y_i|x_n), \ldots, P(y_{N_y}|x_n)\}$ denote the posterior probability distribution over $N_y$ task labels and $P_{ks}(x_n) = \{P(s_1|x_n), \ldots, P(s_i|x_n), \ldots, P(s_{N_s}|x_n)\}$ denote the posterior probability distribution over $N_s$ nuisance labels produced by model $\mathcal{M}_k$ given data instance $x_n$. Ensemble generalization works by stacking the predictions of the base learners in a higher level learning space, where meta learner, denoted as $\tilde{\mathcal{M}}_k$, corrects the predictions of base learners (Wolpert, 1992). Subsequent to training base learners, we assemble the posterior probability vectors of all base learners together: $P_y(x_n) = \{P_{ky}(x_n)\}$ and $P_s(x_n) = \{P_{ks}(x_n)\}$, where $k = 1 : 37$. $\tilde{\mathcal{M}}_k$ is trained using the predictions from all base learners as input attributes: $\tilde{\mathcal{D}}_{\mathrm{train}}^{\mathrm{in}} = \{(P_y(x_n), P_s(x_n))\}$ and correct labels as output: $\tilde{\mathcal{D}}_{\mathrm{train}}^{\mathrm{out}} = \{(y_n, s_n)\}$, where $n = 1 : N_{\mathrm{train}}$. Hold-out $\mathcal{D}_{\mathrm{test}}$ is used to measure the classification performance of both base and meta learners. To make best use of the base learners, we compare the predictive performance of a LR model and a shallow MLP as a meta learner in Table 2.

## A.6    Datasets Description

We used publicly available physiological datasets as well as a benchmark MNIST as follows. The parameters of datasets are also summarized in Table 1.

- **QMNIST:** A hand-written digit image MNIST with extended label information including a writer ID number (Yadav & Bottou, 2019).[3] There are $|S| = 539$ writers for classifying $|Y| = 10$ digits from grayscale $28 \times 28$ pixel images over 60,000 training samples. Additional 297 writers provide 10,000 test samples.

- **Stress:** A physiological dataset considering neurological stress level (Birjandtalab et al., 2016).[4] It consists of multi-modal biosignals for $|Y| = 4$ discrete stress states from $|S| = 20$ healthy subjects, including physical/cognitive/emotional stresses as well as relaxation. The data were collected by $C = 7$ sensors, i.e., electrodermal activity, temperature, three-dimensional acceleration, heart rate, and arterial oxygen level. For each stress status, a corresponding task of 5 minutes long (i.e., $T = 300$ time samples with 1 Hz down-sampling) was assigned to subjects for a total of 4 trials.

---

[2]The $\mathrm{Gumbel}(0, 1)$ distribution can be sampled by drawing $e \sim \mathrm{Exp}(1)$ and computing $g = -\log(e)$.
[3]QMNIST dataset: `https://github.com/facebookresearch/qmnist`
[4]Stress dataset: `https://physionet.org/content/noneeg/1.0.0/`

- **RSVP:** An EEG-based typing interface using rapid serial visual presentation (RSVP) paradigm (Orhan et al., 2012).[5] $|S| = 10$ healthy subjects participated in the experiments at three sessions performed on different days. The dataset consists of 41,400 epochs of $C = 16$ channel EEG data for $T = 128$ samples, which were collected by g.USBamp biosignal amplifier with active electrodes during RSVP keyboard operations. $|Y| = 4$ labels for emotion elicitation, resting-state, or motor imagery/execution task.

- **MI:** The PhysioNet EEG Motor Imagery (MI) dataset (Goldberger et al., 2000).[6] Excluding irregular timestamp, the dataset consists of $|S| = 106$ subjects' EEG data. During the experiments, subjects were instructed to perform cue-based motor execution/imagery tasks while $C = 64$ channels were recorded at a sampling rate of 160 Hz. Focusing on motor imagery tasks, we use the EEG data for three seconds of post-cue interval data (i.e., $T = 480$ time samples). The subject performed $|Y| = 4$-class tasks; either right hand motor imagery, left hand motor imagery, both hands motor imagery, or both feet motor imagery. This resulted in a total of 90 trials per subject.

- **ErrP:** An error-related potential (ErrP) of front-central EEG dataset (Margaux et al., 2012).[7] The dataset consists of EEG data recorded from $|S| = 16$ healthy subjects participating in an offline P300 spelling task, where visual feedback of the inferred letter is provided to the user at the end of each trial for 1.3 seconds to monitor evoked brain responses for erroneous decisions made by the system. EEG data were recorded from $C = 56$ channels for epoched 1.25 seconds at a sampling rate of 200 Hz (i.e., $T = 250$). Across five recording sessions, each subject performed a total of 340 trials. Since it was an offline copy spell task, binary $|Y| = 2$ labels were provided as erroneous or correct feedback.

- **Faces Basic:** An implanted electrocorticography (ECoG) array dataset for visual stimulus experiments (Miller et al., 2015; 2016).[8] ECoG arrays were implanted on the subtemporal cortical surface of $|S| = 14$ epilepsy patients. $|Y| = 2$ classes of grayscale images, either faces or houses, were displayed rapidly in random sequence for 400 ms each with black-screen intervals of 400 ms. The ECoG potentials were measured with respect to a scalp reference and ground, at a sampling rate of 1000 Hz. Subjects performed a basic face and house discrimination task. There were 3 sessions for each patient, with 50 house pictures and 50 face pictures in each run, in total 4,100 samples. We use the first $C = 31$ channels to analyze for $T = 400$. Reusing the public dataset requires the ethics statement information.[9]

- **Faces Noisy:** The implanted ECoG arrays dataset for visual stimulus experiments (Miller et al., 2015; 2017). The experiment is similar to Faces Basic dataset, while pictures of faces and houses are randomly scrambled. There are $|S| = 7$ subjects with $C = 39$ channels. Refer ethics statement to reuse the dataset.[10]

- **ASL:** An EMG dataset for finger gesture identification for American Sign Language (ASL) (Günay et al., 2019).[11] $|S| = 5$ healthy, right-handed, subjects participated in experiments with surface EMG (Delsys Inc. Trigno) recorded at 2 kHz from $|C| = 16$ lower-arm muscles. Subjects shaped their right hand into letters and numbers of the ASL posture set presented as pictures on a computer screen ($|Y| = 33$ postures, 3 trials per

---

[5]RSVP dataset: `http://hdl.handle.net/2047/D20294523`

[6]MI dataset: `https://physionet.org/physiobank/database/eegmmidb/`

[7]ErrP dataset: `https://www.kaggle.com/c/inria-bci-challenge/`

[8]Faces dataset: `https://exhibits.stanford.edu/data/catalog/zk881ps0522`

[9]**Ethics statement**: All patients participated in a purely voluntary manner, after providing informed written consent, under experimental protocols approved by the Institutional Review Board of the University of Washington (#12193). All patient data was anonymized according to IRB protocol, in accordance with HIPAA mandate. These data originally appeared in the manuscript "Spontaneous Decoding of the Timing and Content of Human Object Perception from Cortical Surface Recordings Reveals Complementary Information in the Event-Related Potential and Broadband Spectral Change" published in PLoS Computational Biology in 2016 (Miller et al., 2016).

[10]All patients participated in a purely voluntary manner, after providing informed written consent, under experimental protocols approved by the Institutional Review Board of the University of Washington (#12193). All patient data was anonymized according to IRB protocol, in accordance with HIPAA mandate. These data originally appeared in the manuscript "Face percept formation in human ventral temporal cortex" published in Journal of Neurophysiology in 2017 (Miller et al., 2017).

[11]ASL Dataset: `http://hdl.handle.net/2047/D20294523`

Table 3: DNN model parameters in Fig. 7; $\text{Conv}(h, w)_g^c$ denotes 2D convolution layer with kernel size of $(h, w)$ for output channel of $c$ over group $g$. $\text{FC}(h)$ denotes fully-connected layer with $h$ output nodes. BN denotes batch normalization.

| Classifier $\mathcal{C}$ | Encoder $\mathcal{E}$ | Decoder $\mathcal{D}$ | Nuisance $\mathcal{N}$ | Adversary $\mathcal{A}$ |
|---|---|---|---|---|
| $\text{FC}(2|Z|)$ | $\text{Conv}(1,15)^{50}$ | $\text{FC}(20T)$ | $\text{FC}(2|Z|)$ | $\text{FC}(2|Z|)$ |
| BN+ReLU | BN+ReLU | ReLU | BN+ReLU | BN+ReLU |
| $\text{FC}(|Y|)$ | $\text{Conv}(1,7)^{50}$ | $\text{Conv}(C,1)^{50}$ | $\text{FC}(|S|)$ | $\text{FC}(|S|)$ |
| | BN+ReLU | BN+ReLU | | |
| | $\text{Conv}(1,3)^{50}$ | $\text{Conv}(1,3)^{50}$ | | |
| | BN+ReLU | BN+ReLU | | |
| | $\text{Conv}(C,1)_{50}^{50}$ | $\text{Conv}(1,7)^{50}$ | | |
| | $\text{FC}(|Z|)$ | BN+ReLU | | |
| | | $\text{Conv}(1,15)^{50}$ | | |

posture). Dynamic letters 'J' and 'Z' were omitted, along with the number '0', which is visually the same as the letter 'O'. The participants were given 2 seconds to form the posture, 6 seconds to maintain it, and 2 seconds to rest between trials. The signal is decimated to be $T = 100$.

## A.7 DNN MODEL PARAMETERS

For 2D datasets, we use deep CNN for the encoder $\mathcal{E}$ and decoder $\mathcal{D}$ blocks. For the classifier $\mathcal{C}$, nuisance estimator $\mathcal{N}$, and adversary $\mathcal{A}$, we use a multi-layer perceptron (MLP) having three layers, whose hidden nodes are doubled from the input dimension. We also use batch normalization (BN) and ReLU activation as listed in Table 3. Note that for a tabular data such as Stress datasets, CNN was replaced with 3-layer MLP having ReLU activation and dropout with a ratio of $20\%$. Also the MLP classifier was replaced with CNN for 2D input dimension cases such as in the model A. The number of latent dimensions was chosen $|Z| = 64$. When we need to feed $S$ along with 2D data of $X$ into the CNN encoder such as in the model Ds, dimension mismatch poses a problem. We address this issue by using one linear layer to project $S$ into the temporal dimensional space of $X$ and another linear layer to project it into the spatial dimensional space of $X$. The dot product of those two projected vectors is concatenated as additional channel input. We use $\lambda_* = 0.01$ for the regularization coefficient. We leave hyperparameter exploration to integrate AutoML and AutoBayes as a remaining future work.

## A.8 PERFORMANCE RESULTS

The additional results for the all datasets are listed in Table 4. The results suggest that the best inference strategy highly depends on datasets. Specifically, the best model at one dataset does not perform best for different datasets; e.g., the model non-variational Is was best for ASL dataset, while the model variational Ds was best for RSVP dataset. It suggests that we shall consider different inference strategies for each target dataset and AutoBayes provides such an adaptive framework. Also note that reconstruction loss may not be a good indicator to select the graph model. In addition, a huge performance gap between the best and worst models was observed for some datasets. For example, the task accuracy of $76.4\%$ was achieved with model non-variational Dz for Faces (Noisy) dataset, whereas the model variational B offers $51.4\%$. This implies that we may have a potential risk that one particular model cannot achieve good performance if we do not explore different models.

## A.9 SUBJECT VARIATION PERFORMANCE

For Stress dataset, there are $|S| = 20$ subjects. As we have shown in Fig. 9(a), we demonstrated that AutoBayes can improve robustness against the nuisance variation, i.e., subject ID $S$. In Fig. 10, we show that the task classification accuracy highly depends on the subject ID $S$. Here, the box-whisker plots shows the accuracy distribution over different models from A to Kz. The outliers are identified by a whisker factor of $2.4$ with respective to an inter-quartile range. It is seen that some users (e.g.,

Table 4: Performance of datasets: the reconstruction loss, the scores of nuisance classification and task classification in variational/non-variational and adversarial/non-adversarial setting.

| Dataset | Method | Reconstruction Loss (dB) | | Nuisance Classification (%) | | Task Classification (%) | | Model Complexity | |
|---|---|---|---|---|---|---|---|---|---|
| | | Non-Variational | Variational | Non-Variational | Variational | Non-Variational | Variational | No. of Parameters | Clock Time |
| QMNIST | Model A | $-51.73$ | — | — | — | 99.02 | — | 290K | 01 : 10 : 51 |
| | Model B | $-65.68$ | $-61.62$ | — | — | 98.72 | 99.44 | 978K | 01 : 58 : 51 |
| | Model Cs | $-66.38$ | — | 13.12 | — | 99.32 | — | 3.56M | 01 : 50 : 08 |
| | Model Cy | $-67.74$ | — | 12.17 | — | 99.30 | — | 3.53M | 01 : 48 : 47 |
| | Model Ds | $-57.14$ | $-41.43$ | 10.55 | 9.90 | 99.35 | 99.23 | 3.43M | 01 : 54 : 06 |
| | Model Dz | $-65.04$ | $\mathbf{-66.74}$ | 0.44 | 0.46 | 99.16 | 99.27 | 1.03M | 01 : 21 : 37 |
| | Model Es | $-65.35$ | $-66.56$ | 11.77 | 10.51 | 99.44 | 99.21 | 4.17M | 01 : 55 : 15 |
| | Model Ez | $-65.51$ | $-61.41$ | 2.55 | 14.95 | 99.35 | 99.13 | 4.15M | 02 : 41 : 13 |
| | Model Fs | $-57.39$ | $-43.39$ | 14.94 | 16.50 | 99.34 | 99.40 | 3.49M | 02 : 22 : 03 |
| | Model Fz | $-65.85$ | $-43.42$ | 1.80 | 9.03 | 99.08 | 99.41 | 1.09M | 02 : 55 : 13 |
| | Model Gs | $-64.88$ | $-61.51$ | 9.78 | 10.25 | 98.54 | 98.88 | 4.23M | 01 : 53 : 51 |
| | Model Gz | $-65.68$ | $-42.05$ | 9.71 | 12.36 | 99.12 | 98.73 | 4.20M | 01 : 58 : 44 |
| | Model Hs | $-66.02$ | $-43.32$ | 15.94 | 16.56 | 99.18 | 99.39 | 3.49M | 02 : 24 : 17 |
| | Model Hz | $-65.85$ | $-43.45$ | 13.20 | 14.70 | 99.47 | 99.28 | 3.49M | 02 : 24 : 37 |
| | Model Is | $-65.35$ | $-45.41$ | **15.96** | **18.57** | 99.46 | 99.32 | 4.28M | 02 : 26 : 26 |
| | Model Iz | $-65.84$ | $-45.46$ | 14.97 | 15.45 | **99.54** | 99.28 | 4.28M | 02 : 23 : 56 |
| | Model Js | $-59.02$ | $-57.3$ | 11.41 | 11.21 | 99.47 | 99.39 | 4.11M | 02 : 19 : 28 |
| | Model Jz | $-67.96$ | $-61.51$ | 6.44 | 5.02 | 98.85 | **99.46** | 1.71M | 02 : 14 : 30 |
| | Model Ks | $-65.51$ | $-63.35$ | 11.59 | 1.16 | 99.49 | 99.10 | 4.12M | 02 : 21 : 54 |
| | Model Kz | $\mathbf{-67.33}$ | $-61.20$ | 6.32 | 6.94 | 99.15 | 99.15 | 1.71M | 02 : 14 : 24 |
| Stress | Model A | $-56.31$ | — | — | — | 85.87 | — | 32.7K | 00 : 00 : 35 |
| | Model B | $-66.56$ | $-59.41$ | — | — | 94.79 | 92.67 | 97.0K | 00 : 01 : 32 |
| | Model Cs | $-67.74$ | — | 59.46 | — | 93.48 | — | 50.0K | 00 : 00 : 50 |
| | Model Cy | $-66.56$ | — | 75.77 | — | 91.93 | — | 48.0K | 00 : 00 : 55 |
| | Model Ds | $-61.94$ | $-36.04$ | 59.90 | 28.37 | 93.26 | 83.70 | 95.3K | 00 : 01 : 02 |
| | Model Dz | $-66.02$ | $-48.40$ | 81.17 | 36.21 | 94.22 | 79.76 | 99.0K | 00 : 01 : 03 |
| | Model Es | $-66.38$ | $-63.35$ | 54.21 | 79.76 | 94.00 | 92.05 | 95.3K | 00 : 01 : 08 |
| | Model Ez | $-64.73$ | $-59.25$ | **90.35** | 91.92 | 95.02 | 30.00 | 99.7K | 00 : 01 : 46 |
| | Model Fs | $-64.73$ | $-38.68$ | 68.45 | 40.74 | 94.07 | 87.80 | 94.4K | 00 : 01 : 04 |
| | Model Fz | $-66.94$ | $-38.57$ | 83.25 | 5.18 | 94.92 | 87.24 | 98.1K | 00 : 01 : 40 |
| | Model Gs | $-67.96$ | $\mathbf{-64.73}$ | 53.94 | 25.88 | 93.61 | 86.56 | 97.3K | 00 : 01 : 11 |
| | Model Gz | $-65.85$ | $-39.16$ | 82.86 | 69.26 | 94.11 | 89.04 | 102K | 00 : 01 : 01 |
| | Model Hs | $-65.04$ | $-38.47$ | 78.36 | 72.42 | 94.72 | **92.86** | 94.4K | 00 : 01 : 04 |
| | Model Hz | $-66.38$ | $-38.37$ | 84.10 | 71.07 | 94.57 | 90.73 | 101K | 00 : 01 : 06 |
| | Model Is | $-66.74$ | $-47.94$ | 79.51 | 74.38 | 94.74 | 91.94 | 96.4K | 00 : 01 : 04 |
| | Model Iz | $-67.96$ | $-47.98$ | 84.46 | 68.63 | 94.80 | 90.52 | 103K | 00 : 01 : 04 |
| | Model Js | $-67.13$ | $-36.17$ | 79.36 | **92.47** | **95.35** | 30.00 | 140K | 00 : 01 : 21 |
| | Model Jz | $-66.74$ | $-54.02$ | 86.27 | 58.59 | 95.17 | 86.99 | 135K | 00 : 02 : 07 |
| | Model Ks | $\mathbf{-68.64}$ | $-51.50$ | 73.57 | 87.33 | 94.65 | 86.74 | 146K | 00 : 01 : 20 |
| | Model Kz | $-66.56$ | $-51.94$ | 85.00 | 61.84 | 94.35 | 86.34 | 141K | 00 : 02 : 05 |
| RSVP | Model A | $-30.69$ | — | — | — | 93.07 | — | 268K | 00 : 48 : 25 |
| | Model B | $-34.27$ | $-35.36$ | — | — | 93.06 | 91.89 | 1.87M | 01 : 00 : 35 |
| | Model Cs | $-31.33$ | — | 90.12 | — | 91.56 | — | 437K | 00 : 55 : 35 |
| | Model Cy | $-31.57$ | — | 90.38 | — | 91.54 | — | 435K | 00 : 54 : 29 |
| | Model Ds | $-35.61$ | $-30.17$ | 91.33 | 84.77 | 91.16 | **93.42** | 2.01M | 00 : 56 : 05 |
| | Model Dz | $-35.27$ | $-35.37$ | 92.42 | 86.84 | 92.44 | 92.71 | 1.87M | 00 : 48 : 35 |
| | Model Es | $-35.61$ | $-31.44$ | 91.74 | 90.46 | **93.23** | 92.99 | 2.02M | 00 : 54 : 43 |
| | Model Ez | $-35.62$ | $-35.52$ | **94.26** | **93.01** | 92.65 | 91.99 | 2.03M | 01 : 16 : 32 |
| | Model Fs | $-35.60$ | $-30.17$ | 91.03 | 90.38 | 92.15 | 93.27 | 2.06M | 01 : 00 : 52 |
| | Model Fz | $-32.94$ | $-30.16$ | 9.57 | 9.88 | 90.21 | 91.04 | 1.93M | 01 : 08 : 24 |
| | Model Gs | $-35.78$ | $-31.24$ | 92.17 | 92.90 | 89.83 | 86.82 | 2.03M | 00 : 57 : 25 |
| | Model Gz | $-35.28$ | $-30.34$ | 91.27 | 90.18 | 92.15 | 91.31 | 2.03M | 00 : 52 : 22 |
| | Model Hs | $-35.40$ | $-30.18$ | 93.89 | 91.31 | 93.05 | 91.22 | 2.06M | 01 : 04 : 10 |
| | Model Hz | $-35.39$ | $-30.18$ | 91.49 | 89.84 | 92.65 | 92.76 | 2.06M | 01 : 04 : 20 |
| | Model Is | $-35.37$ | $-30.35$ | 93.37 | 90.32 | 92.94 | 91.60 | 2.08M | 01 : 04 : 16 |
| | Model Iz | $-35.37$ | $-30.36$ | 91.36 | 90.96 | 91.41 | 91.92 | 2.08M | 01 : 00 : 53 |
| | Model Js | $\mathbf{-36.10}$ | $-36.09$ | 92.78 | 9.92 | 90.82 | 92.74 | 3.64M | 01 : 02 : 55 |
| | Model Jz | $-35.82$ | $\mathbf{-36.65}$ | 93.60 | 82.62 | 93.12 | 92.85 | 3.49M | 01 : 01 : 37 |
| | Model Ks | $-35.65$ | $-36.05$ | 90.93 | 92.86 | 93.19 | 90.54 | 3.65M | 01 : 01 : 11 |
| | Model Kz | $-35.53$ | $-36.01$ | 91.99 | 82.10 | 92.81 | 93.03 | 3.50M | 00 : 58 : 04 |

Table 4: Performance of datasets (continued)

| Dataset | Method | Reconstruction Loss (dB) | | Nuisance Classification (%) | | Task Classification (%) | | Model Complexity | |
|---|---|---|---|---|---|---|---|---|---|
| | | Non-Variational | Variational | Non-Variational | Variational | Non-Variational | Variational | No. of Parameters | Clock Time |
| MI | Model A | $-30.28$ | — | — | — | 55.85 | — | 454K | $02:47:39$ |
| | Model B | $-32.17$ | $-32.24$ | — | — | 56.32 | 47.61 | 6.29M | $03:53:06$ |
| | Model Cs | $-32.12$ | — | 35.99 | — | 52.65 | — | 5.89M | $03:35:09$ |
| | Model Cy | $-32.15$ | — | 43.60 | — | 52.98 | — | 5.84M | $03:34:53$ |
| | Model Ds | $-31.34$ | $-20.20$ | 74.15 | 1.14 | 24.26 | 24.89 | 10.9M | $03:25:51$ |
| | Model Dz | $-32.14$ | $-35.92$ | 4.82 | 9.01 | 55.26 | 51.80 | 6.30M | $02:57:19$ |
| | Model Es | $-32.22$ | $-30.90$ | 61.95 | 0.74 | 44.74 | 24.85 | 11.7M | $03:25:11$ |
| | Model Ez | $-32.52$ | $-30.82$ | 5.77 | 8.21 | 54.12 | 48.65 | 11.7M | $04:43:56$ |
| | Model Fs | $-30.36$ | $-20.35$ | 38.60 | 0.66 | 48.90 | 51.91 | 11.2M | $04:16:05$ |
| | Model Fz | $-31.86$ | $-29.77$ | 3.05 | 0.96 | 57.83 | 25.40 | 6.54M | $04:24:11$ |
| | Model Gs | $-32.16$ | $-30.07$ | 33.97 | 0.55 | 53.01 | 24.82 | 11.7M | $03:30:57$ |
| | Model Gz | $-32.31$ | $-30.06$ | 4.82 | 0.96 | 52.61 | 26.40 | 11.7M | $03:30:49$ |
| | Model Hs | $-32.11$ | $-30.08$ | 88.42 | 57.87 | 52.68 | 49.04 | 11.2M | $04:15:27$ |
| | Model Hz | $-31.99$ | $-30.02$ | 43.93 | 1.07 | 57.21 | 25.96 | 11.2M | $04:15:56$ |
| | Model Is | $-32.27$ | $-30.08$ | 85.55 | 54.99 | 55.00 | 24.26 | 12.0M | $04:00:46$ |
| | Model Iz | $-32.35$ | $-30.09$ | 48.49 | 1.03 | 53.57 | 26.03 | 12.0M | $03:59:26$ |
| | Model Js | $-30.29$ | $-30.10$ | 49.19 | 0.80 | 41.54 | 24.93 | 17.0M | $04:01:20$ |
| | Model Jz | $-32.88$ | $-35.14$ | 43.64 | 31.10 | 57.50 | 44.93 | 12.3M | $03:59:00$ |
| | Model Ks | $-30.79$ | $-30.18$ | 81.18 | 0.77 | 23.79 | 25.18 | 17.0M | $04:00:59$ |
| | Model Kz | $-32.27$ | $-32.44$ | 29.26 | 28.31 | 48.12 | 48.79 | 12.3M | $03:43:01$ |
| ErrP | Model A | $-31.04$ | — | — | — | 69.89 | — | 301K | $00:48:18$ |
| | Model B | $-41.26$ | $-39.79$ | — | — | 71.81 | 71.39 | 3.40M | $01:05:07$ |
| | Model Cs | $-39.26$ | — | 94.95 | — | 63.68 | — | 1.05M | $00:56:24$ |
| | Model Cy | $-41.51$ | — | 98.98 | — | 70.07 | — | 1.05M | $00:59:07$ |
| | Model Ds | $-39.44$ | $-29.92$ | 98.68 | 7.69 | 69.11 | 69.77 | 4.04M | $00:56:40$ |
| | Model Dz | $-42.52$ | $-39.46$ | 97.30 | 68.93 | 68.09 | 75.91 | 3.41M | $00:47:42$ |
| | Model Es | $-39.49$ | $-38.91$ | 97.12 | 92.91 | 70.01 | 65.38 | 4.14M | $00:59:27$ |
| | Model Ez | $-41.17$ | $-41.98$ | 47.18 | 99.64 | 70.91 | 72.42 | 4.15M | $01:18:58$ |
| | Model Fs | $-39.54$ | $-30.00$ | 98.32 | 6.73 | 71.45 | 70.07 | 4.13M | $01:09:08$ |
| | Model Fz | $-41.35$ | $-30.10$ | 93.33 | 8.35 | 66.71 | 70.19 | 3.50M | $01:17:28$ |
| | Model Gs | $-40.23$ | $-33.96$ | 97.00 | 0.42 | 70.85 | 70.31 | 4.14M | $00:56:57$ |
| | Model Gz | $-41.02$ | $-29.94$ | 96.57 | 98.68 | 69.23 | 67.31 | 4.15M | $00:57:12$ |
| | Model Hs | $-40.03$ | $-28.32$ | 98.14 | 98.02 | 67.85 | 29.93 | 4.13M | $01:10:47$ |
| | Model Hz | $-41.19$ | $-29.90$ | 96.81 | 97.12 | 68.81 | 69.11 | 4.13M | $01:05:37$ |
| | Model Is | $-38.09$ | $-30.07$ | 98.26 | 96.33 | 59.62 | 67.31 | 4.23M | $01:07:48$ |
| | Model Iz | $-40.54$ | $-29.99$ | 96.21 | 96.33 | 70.25 | 66.95 | 4.23M | $01:10:42$ |
| | Model Js | $-40.33$ | $-34.44$ | 98.20 | 6.07 | 68.57 | 68.03 | 7.21M | $01:11:08$ |
| | Model Jz | $-42.40$ | $-41.27$ | 99.04 | 95.13 | 72.54 | 69.29 | 6.54M | $01:06:01$ |
| | Model Ks | $-38.85$ | $-37.71$ | 98.86 | 5.77 | 68.63 | 69.29 | 7.22M | $01:09:38$ |
| | Model Kz | $-42.48$ | $-40.05$ | 98.32 | 95.01 | 72.36 | 69.65 | 6.55M | $01:05:53$ |

Table 4: Performance of datasets (continued)

| Dataset | Method | Reconstruction Loss (dB) | | Nuisance Classification (%) | | Task Classification (%) | | Model Complexity | |
|---|---|---|---|---|---|---|---|---|---|
| | | Non-Variational | Variational | Non-Variational | Variational | Non-Variational | Variational | No. of Parameters | Clock Time |
| Faces Basic | Model A | −29.95 | — | — | — | 63.30 | — | 332K | 00 : 40 : 22 |
| | Model B | −33.68 | −30.10 | — | — | 48.56 | 51.12 | 5.27M | 00 : 57 : 51 |
| | Model Cs | −32.18 | — | 80.45 | — | 64.50 | — | 960K | 00 : 46 : 11 |
| | Model Cy | −32.96 | — | 87.26 | — | 65.62 | — | 954K | 00 : 46 : 07 |
| | Model Ds | −32.99 | −30.10 | 92.23 | 7.69 | 62.74 | 48.08 | 5.80M | 00 : 48 : 29 |
| | Model Dz | −31.68 | −23.37 | 88.70 | 7.77 | 66.99 | 49.28 | 5.28M | 00 : 40 : 52 |
| | Model Es | −31.98 | −30.08 | 92.95 | 6.73 | 50.96 | 53.12 | 5.88M | 00 : 47 : 36 |
| | Model Ez | −31.84 | −30.03 | 38.94 | 97.60 | 50.96 | 51.36 | 5.91M | 01 : 05 : 37 |
| | Model Fs | −33.32 | −30.11 | 96.07 | 8.09 | 61.14 | 62.82 | 5.93M | 00 : 54 : 31 |
| | Model Fz | −32.95 | −28.80 | 49.60 | 10.02 | 61.30 | 61.14 | 5.40M | 01 : 06 : 39 |
| | Model Gs | −32.56 | −29.76 | 91.11 | 7.05 | 63.38 | 49.92 | 5.89M | 00 : 46 : 34 |
| | Model Gz | −33.13 | −30.11 | 85.02 | 83.41 | 63.86 | 64.02 | 5.91M | 00 : 46 : 19 |
| | Model Hs | −32.03 | −30.08 | 98.00 | 86.22 | 61.14 | 64.42 | 5.93M | 00 : 51 : 52 |
| | Model Hz | −33.29 | −29.41 | 91.11 | 83.81 | 65.46 | 61.94 | 5.93M | 00 : 54 : 32 |
| | Model Is | −31.63 | −30.11 | 97.92 | 94.39 | 62.34 | 61.94 | 6.01M | 00 : 54 : 30 |
| | Model Iz | −33.20 | −30.06 | 91.67 | 89.10 | 63.94 | 67.31 | 6.01M | 00 : 51 : 56 |
| | Model Js | −33.28 | −30.12 | 94.87 | 8.33 | 51.04 | 52.23 | 10.9M | 00 : 53 : 17 |
| | Model Jz | −32.21 | −29.50 | 93.83 | 7.29 | 65.79 | 51.28 | 10.3M | 00 : 56 : 36 |
| | Model Ks | −31.12 | −29.88 | 88.94 | 7.45 | 51.92 | 53.85 | 10.9M | 00 : 55 : 41 |
| | Model Kz | −32.69 | −30.09 | 93.43 | 7.93 | 51.76 | 51.84 | 10.3M | 00 : 56 : 37 |
| Faces Noisy | Model A | −30.09 | — | — | — | 75.94 | — | 333K | 00 : 24 : 12 |
| | Model B | −30.35 | −30.09 | — | — | 73.59 | 51.41 | 5.27M | 00 : 33 : 07 |
| | Model Cs | −30.10 | — | 95.62 | — | 75.16 | — | 664K | 00 : 30 : 04 |
| | Model Cy | −30.56 | — | 96.56 | — | 71.56 | — | 662K | 00 : 27 : 47 |
| | Model Ds | −30.22 | −27.90 | 82.34 | 13.28 | 74.84 | 51.72 | 5.55M | 00 : 27 : 45 |
| | Model Dz | −30.11 | −30.09 | 96.09 | 14.38 | 76.41 | 53.91 | 5.28M | 00 : 24 : 18 |
| | Model Es | −30.09 | −28.70 | 91.09 | 13.28 | 74.38 | 52.50 | 5.59M | 00 : 27 : 45 |
| | Model Ez | −30.47 | −28.58 | 21.41 | 93.75 | 70.94 | 52.97 | 5.61M | 00 : 40 : 16 |
| | Model Fs | −30.14 | −30.08 | 95.62 | 13.75 | 71.88 | 75.62 | 5.68M | 00 : 32 : 51 |
| | Model Fz | −29.96 | −27.76 | 27.50 | 17.03 | 72.50 | 72.19 | 5.40M | 00 : 40 : 20 |
| | Model Gs | −28.46 | −30.15 | 93.75 | 13.91 | 71.56 | 52.50 | 5.59M | 00 : 30 : 07 |
| | Model Gz | −30.59 | −30.09 | 94.53 | 80.94 | 75.00 | 75.16 | 5.61M | 00 : 27 : 52 |
| | Model Hs | −30.04 | −30.08 | 98.49 | 88.59 | 75.59 | 69.06 | 5.68M | 00 : 31 : 14 |
| | Model Hz | −30.30 | −30.06 | 95.94 | 91.09 | 75.47 | 76.09 | 5.68M | 00 : 32 : 58 |
| | Model Is | −30.10 | −30.04 | 97.97 | 96.88 | 68.91 | 69.53 | 5.72M | 00 : 31 : 27 |
| | Model Iz | −30.62 | −29.86 | 88.91 | 87.19 | 74.06 | 72.50 | 5.72M | 00 : 33 : 43 |
| | Model Js | −30.08 | −28.72 | 95.69 | 15.94 | 65.31 | 53.59 | 10.6M | 00 : 33 : 16 |
| | Model Jz | −30.57 | −30.03 | 96.62 | 14.22 | 71.56 | 52.66 | 10.3M | 00 : 35 : 01 |
| | Model Ks | −30.29 | −30.14 | 65.62 | 15.52 | 54.06 | 53.44 | 10.6M | 00 : 31 : 34 |
| | Model Kz | −30.12 | −28.45 | 94.84 | 12.66 | 76.56 | 54.23 | 10.3M | 00 : 34 : 54 |
| ASL | Model A | −24.22 | — | — | — | 41.69 | — | 588K | 01 : 18 : 06 |
| | Model B | −23.89 | −24.08 | — | — | 3.03 | 37.80 | 1.53M | 01 : 34 : 55 |
| | Model Cs | −24.07 | — | 93.63 | — | 38.35 | — | 726K | 01 : 26 : 27 |
| | Model Cy | −24.14 | — | 94.63 | — | 38.28 | — | 729K | 01 : 26 : 31 |
| | Model Ds | −24.07 | −24.08 | 93.74 | 94.29 | 39.23 | 41.32 | 1.63M | 01 : 32 : 02 |
| | Model Dz | −24.47 | −24.69 | 95.99 | 95.10 | 43.83 | 40.89 | 1.53M | 01 : 16 : 50 |
| | Model Es | −24.07 | −24.07 | 94.00 | 93.60 | 40.07 | 40.38 | 1.65M | 01 : 32 : 04 |
| | Model Ez | −24.96 | −24.10 | 43.16 | 85.45 | 43.56 | 37.23 | 1.65M | 01 : 55 : 58 |
| | Model Fs | −24.07 | −24.08 | 93.93 | 97.58 | 38.75 | 42.27 | 2.00M | 01 : 39 : 40 |
| | Model Fz | −24.08 | −24.08 | 9.99 | 10.79 | 28.25 | 42.16 | 1.89M | 01 : 50 : 56 |
| | Model Gs | −24.07 | −24.08 | 94.45 | 93.81 | 38.81 | 39.83 | 1.65M | 01 : 29 : 42 |
| | Model Gz | −24.50 | −24.81 | 95.69 | 94.76 | 47.43 | 43.32 | 1.65M | 01 : 27 : 01 |
| | Model Hs | −25.10 | −24.08 | 96.61 | 94.26 | 49.30 | 36.39 | 2.00M | 01 : 39 : 54 |
| | Model Hz | −24.87 | −24.08 | 94.77 | 94.20 | 48.31 | 37.33 | 2.00M | 01 : 45 : 32 |
| | Model Is | −24.87 | −24.08 | 96.54 | 94.37 | 51.12 | 38.31 | 2.01M | 01 : 39 : 47 |
| | Model Iz | −24.74 | −25.03 | 95.81 | 93.98 | 49.47 | 38.45 | 2.01M | 01 : 45 : 43 |
| | Model Js | −24.07 | −24.11 | 93.64 | 97.09 | 38.39 | 36.77 | 2.92M | 01 : 47 : 38 |
| | Model Jz | −24.09 | −24.11 | 14.27 | 96.44 | 6.24 | 37.25 | 2.79M | 01 : 35 : 45 |
| | Model Ks | −24.11 | −24.05 | 93.10 | 16.26 | 38.07 | 8.19 | 2.93M | 01 : 39 : 54 |
| | Model Kz | −24.22 | −24.22 | 12.34 | 95.83 | 3.03 | 37.75 | 2.80M | 01 : 40 : 42 |

$S = 8$) have superior performance whereas classification task is harder for some other users (e.g., $S = 6$). Our AutoBayes can well resolve the issues of such a nuisance variation by linking the adversarial block for $S$-independent latent variables $Z$ to generate subject-invariant feature.

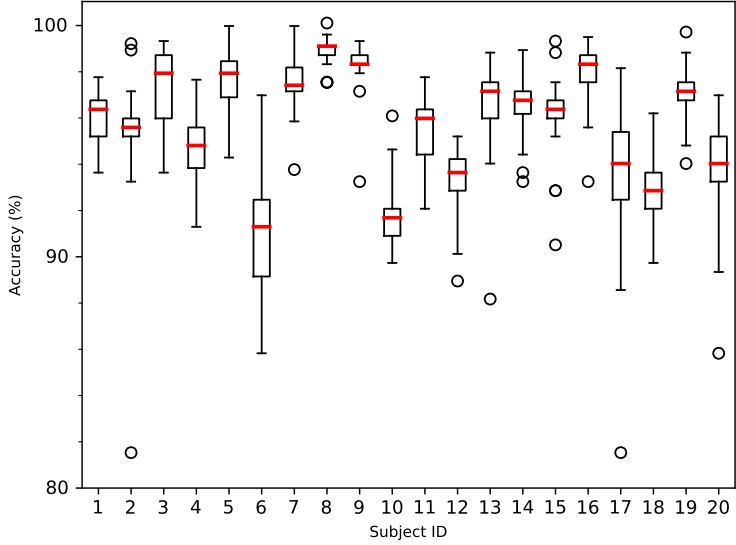

Figure 10: Task classification accuracy across subject ID for Stress dataset.

## A.10 TIME COMPLEXITY ANALYSIS

In Fig. 9(b), we have shown the accuracy vs. the space complexity. In this section, we evaluate the time complexity in Figs. 11(a) and (b), which show the task classification accuracy as a function of computation time for training and testing, respectively, for the Stress dataset. As in the same setting of Fig. 9(b), we explored different DNN configurations for the models A, B, and Js, by sweeping the number of hidden layers and hidden nodes. Some Pareto-front DNN configurations having lower complexity and higher accuracy are connected with lines. We used pytorch on NVIDIA Tesla K80 GPU with CUDA 10.1. It is seen that the standard classifier model A outperforms the other models in lower complexity regimes, whereas our AutoBayes can achieve better Pareto front for higher accuracy regimes. It should be also noted that the increase of the time complexity is not so significant (by a few folds) in comparison to that of the space complexity (by a few magnitudes) in Fig. 9(b).

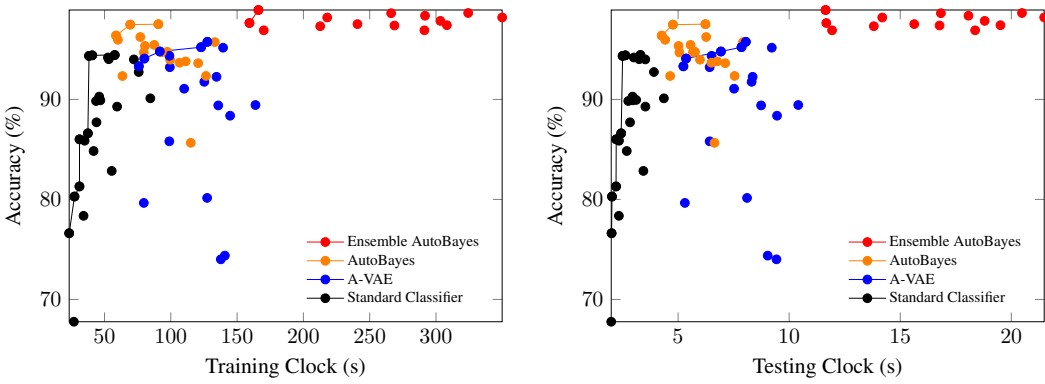

Figure 11: Task classification accuracy as a function of time complexity for Stress dataset.

