# OpenReview forum: "AutoBayes: Automated Bayesian Graph Exploration for Nuisance-Robust Inference"
_ICLR.cc/2021/Conference — Reject_

### Official Review · AnonReviewer3 · 2020-10-26
**Interesting idea, encouraging results,  but poorly organized paper**

**Rating:** 4
**Confidence:** 3

**Review:**

The authors present a novel method dubbed AutoBayes that tries to find optimal Bayesian graph models for "nuisance-robust" deep learning. They employ the Bayes-Ball algorithm to construct reasonable inference graphs from a generative model given by iterative search. The corresponding DNN modules are then built/linked and trained using a form of variational inference with adversarial regularization where applicable. The authors also propose the use of an ensembling approach to further improve robustness of the "best" model.

Pros:
- Interesting and novel approach
- Experimental results are encouraging and seem to demonstrate a clear advantage of AutoBayes over other methods

Cons:
- The organization of the paper is erratic; overall structure doesn't flow well.
- Multiple figures have details (e.g. color or style of arrows, shading, etc.) which are never explained.
- Little to no context is given for what nuisance-robust learning (and correspondingly, nuisance "variation" variables) actually is. Similarly for the datasets and baselines.
- No discussion of time/space complexity or computational demands, which is surprising given the apparent combinatorial complexity of the nested for-loops in Algorithm 1.
- Placement with respect to existing work is somewhat vague (the authors refer to "similarities" and "relationships" but do not concretely describe them).
- The paper needs to be proof read several more times. There are numerous grammatical errors and poorly phrased sentences. I will give a few examples below, but this is largely up to the authors to sort out.

Overall, the paper reads more like a rough draft of a technical report than a standalone research article. It's confusing, difficult to read (I found that I needed to jump around and re-read a lot in order to understand what the authors were trying to say), and fails to give necessary context to readers who aren't familiar with a very specific subset of the deep learning literature. While it's of course always fine to defer to references for details, it's important that the reader can broadly understand your method and the problem it's trying to solve without hunting down a dozen references. This paper seems to assume that the reader has read every relevant paper and is arriving at this work in sequence.

If the authors are willing to do some reorganizing and more diligent proof-reading, I'd be happy to reconsider my rating after seeing the revisions.

A couple of examples of poorly phrased or grammatically incorrect sentences:

"It may be because the probabilistic ~relation~ *relationships* *in the* underlying data ~varies~ vary ~over~ across datasets" (pg 2)

"The ~whole~ DNN blocks are trained with ~adversary~ *adversarial* learning ~in a~ *using* variational Bayesian inference." (pg.3)

---

> ### Author Response · Authors · 2020-11-24
> **Authors' Response to AnonReviewer3**
>
> We thank the reviewer for reviewing our paper with detailed suggestions. We are glad to hear that the reviewer found our paper interesting and encouraging. Although there remain unsolved challenges to be dealt with in the future work, we believe that our paper provides sufficiently novel and useful ideas for the research community. Please find our responses below for respective queries:
> 1. On style: We made more diligent proofreading, revised for any grammatical errors and word choice, and provided additional explanations about shading in the caption of Fig. 3, thick circles and different line colors in the captions of Figs. 5 and 6. Because other reviewers found that the paper was well written and organized, we focused on revisions to add more discussions for better flows rather than a drastic change of the paper organization. We made major revision to improve readability under a hard limitation of 9-page length. Nevertheless, we admit that the revised paper still requires some preliminary knowledge for reading comprehensively. We hope our revision made it sufficient for a wider range of readers.
> 2. On context: We admit that we should have explained better the context of nuisance robustness. In response to Reviewer3, we clarified the concept of nuisance robustness in Section 1. One of the major hurdles in analyzing physiological datasets is the change in data distributions across different subjects and their biological state during recording sessions. There is a vast amount of research to discover subject-invariant features relevant for task classification to obviate the need for frequent calibration required in human-machine interfaces. AutoBayes is an AutoML framework targeted to provide robustness against nuisance variables, such as subject identities and recording sessions in the classification of physiological datasets. The nuisance-robust machine learning is also useful for various other applications. For example of image recognition, some factors such as image sensor specs, ambient environmental conditions, photographer's identity may indirectly affect the performance. For speech recognitions, many nuisance factors such as speaker's attributes, microphone's condition, recoding space etc. may change the classification accuracy. Those inherent nuisance factors are modeled with a random variable $S$ in Bayesian graph. We further added new experimental results in Figs. 9(a) and 10 to explain the nuisance robustness. Please refer our response 1 to Reviewer 4.
> 3. On time/space complexity: We agree with the reviewer that time and space complexity analysis would be a necessary part of discussion in an automated machine learning research. In response to Reviewer 3, we expanded the discussion -- see Fig. 9(b) --, where we added a detailed comparison of the accuracy of different methods vs. the number of trainable parameters. The results well supported that the AutoBayes can outperform individual model in Pareto sense for a fear comparison at the same complexity although it is difficult to achieve lower complexity. In addition, we added measurement results of wall-clock runtime in Table 4. We also evaluated the time complexity for different network size in Figs. 11(a) and (b), which show the task classification accuracy as a function of computation time for training and testing, respectively, for the Stress dataset. We accordingly added discussion for those new results in the revised paper. Please refer our responses 1 and 3 to Reviewer 2, response 2 to Reviewer 4, and response 3 to Reviewer 1.
> 4. On baselines and datasets: We added more detailed description of related works in Sections 2 and A.1. We added more detailed description of datasets in Section A.6. Due to the page limitation, we decided to keep concise descriptions in the main body and to move detailed descriptions in appendix.
>
> Please do not hesitate to post further comments or questions. As our paper was greatly improved with additional explanations, discussions and results, we hope you would re-assess our paper with higher rating.

---

### Official Review · AnonReviewer1 · 2020-10-27
**Very interesting idea and results, but relevant previous work regarding Bayesian networks and structure learning is not discussed at all.**

**Rating:** 5
**Confidence:** 4

**Review:**

The authors propose a framework, called AutoBayes, to automatically
detect the conditional relationship between data features (X), task
labels (Y), nuisance variation labels (S), and potential latent
variables (Z) in DNN architectures.  Assuming a Bayesian network (BN)
which represents the (possibly) conditional independencies between the
aforementioned variables, the authors propose a learning algorithm
which consists of applying Bayes-ball to detect and prune unnecessary
edges in the graph (effectively finding a subgraph, independence map
of the BN), train the resulting DNN architecture, and choose the network
which achieves the highest validation performance.  This idea is
interesting, especially compared to hyperparameter optimization
approaches for model tuning, and the results seem convincing.

However, relevant previous work is not cited and discussed in the paper.
Specifically, BN structure learning and inference in BNs (both of
which are well studied and have extensive literature) are fully
relevant, but are not discussed or mentioned at all.  For instance,
the paper uses undefined terms such as "Bayesian graph model," "Bayesian
graphs," and "graph model," in place of Bayesian network (which is
rigorously defined).  It is important that such related previous work
be discussed to delineate what is novel in the presented approach and
place its contributions within the greater context of this previous
work.  This inclusion would also help the presentation of concepts in
the paper.  For instance, the discussion surrounding equations 1
and 2, i.e.:
"The chain rule can yield the following factorization for a generative
model from Y to X (note that at most 4! factorization orders exist
including useless ones such as reverse direction from X to Y )...," is
the concept of an elimination ordering in the elimination
algorithm for BNs (and graphical models in general).  Showcasing the
presented work in this light, (i.e., as a natural
combination of BN structure learning with macro-level neural-architecture
optimization) would be particularly novel and compelling.

Finally, it is important to discuss the complexity of the presented
algorithm.  Given the Bayesian networks (BNs) in Algorithm 1,  each
independence map (and the underlying DNN
architecture) must be trained then validated.  This algorithm scales
factorially in the number of nodes in the BN.  It is great that the
selected subgraph performs so well (Figure 2), but super-exponential
complexity multiplied by DNN cross-validation training is going to be
very hard to do as m and n grow in Algorithm 1.

Other comments:

-The definition of nuisance and nuisance-variables are implicitly
assumed throughout the paper.  An exact definition of what is meant by
nuisance, in the context, of the paper would be very helpful.

-Algorithm 1 is mostly one large block of text, and is very hard to
parse on the first read.

-In the main contributions enumerated from pages 2-3, contribution 3
looks redundant given contribution 1.

-"Besides fully-supervised training, AutoBayes can automatically build some relevant graphi-
cal models suited for semi-supervised learning." <- please include a
link to where this is discussed.  The enumerated list of contributions
would be a perfect roadmap for the paper (just include references to
sections after every contribution)

-"We note that this paper relates to some existing literature... as addressed in Appendix A.1. Nonetheless, AutoBayes
is a novel framework that diverges from AutoML, which is mostly employed to architecture tuning
at a micro level.  Our work focuses on exploring neural architectures at a macro level, which is not
an arbitrary diversion, but a necessary interlude." <- Appendix A.1
should really be included in the
paper, related work is a not an optional section.  For instance, the
reader may not know what the authors mean in terms of micro versus
macro level.  Reading this without further explanation until later
in the paper, it would
seem that micro-level is more nuanced than macro-level and the former
perhaps subsumes the latter; the authors should detail what they mean
and distinguish their work from previous works in the main paper.

-The terminology is very clumsy: "Bayesian graph models," "Bayesian
graph," and "graph model" are not established terms and, as such,
should be defined so the reader knows what type of ML method is being discussed.  The authors should specify that they have a Bayesian
network whose factorization describes the conditional relationship
between (random) variables.

-"VAE Evidence Lower Bound (ELBO) concept" <- please include citation
for the ELBO

-"How to handle the exponentially growing search space of possible
Bayesian graphs along with the number of random variables remains a
challenging future work." <- this is exactly structure learning in
Bayesian networks (see the Bayesian information criterion, i.e., BIC score).

---

> ### Author Response · Authors · 2020-11-25
> **Authors' Response to AnonReviewer1**
>
> We appreciate you for your careful reviews and suggestions. We are happy that you found our paper interesting and convincing.  We made a major revision to improve the paper. Although we admit that many future works remain to rigorously demonstrate the usefulness of our idea, we believe that our proposed framework has some important insides and contributions for the research community. We hope you would reconsider assessing higher rating for the revised paper. Detailed responses are summarized below:
> 1. On related work: We thank you for the important comment. Accordingly, we added additional literature on Bayesian network and structure learning. We certainly admit that some terminologies (graph model etc.) are not well defined. As pointed out, our Bayesian graph should be identical to Bayesian network. Considering the fact that Bayesian network is also confusing as it may refer some specific deep neural network instances having Bayesian inference rather than mathematical concepts, we decided to keep using Bayesian graph. Instead, we added a sentence clarifying that our Bayesian graph is same as Bayesian network. There is another reason why we still use undefined graph concepts; besides Bayesian network (joint probability factoring), we also explicitly represent inference strategy in graphical model as in Fig. 5 (for conditional probability factoring) as there any non-unique multiple strategies. We believe it is not a major issue.
> 2. On novelty: As you described, our idea has a solid set of novelties over existing work on Bayesian network and structure learning. Besides the novelties you listed, one of major contributions includes the fact that AutoBayes can reasonably involve adversarial censoring for latent variables which is independent of another factor in a systematic way. In addition, we believe that the introduction of ensemble stacking in AutoML framework is novel and advantageous as most hyperparameter design methods throw away any weaker base models explored. Our results of ensemble AutoBayes showed a significant gain empirically.
> 3. On complexity: We completely agree with you that the near-exhaustive search of our AutoBayes algorithm has a complexity issue when the number of nodes is large. Nevertheless, we believe that the required number of nodes to model realistic datasets is limited. Please refer our response 1 to Reviewer2. In addition to the discussion of node size, we added discussion and new experimental results to show the trade-off between accuracy and complexity; specifically, Fig. 9(b) and Fig. 11. We empirically showed that our AutoBayes can outperform individual models when comparing at the same complexity. We believe those revisions made our paper improved a lot.
> 4. We added more clarification of what is nuisance. And, we added a new result in Fig. 9(a) and Fig. 10 to show the benefit of AutoBayes for minimizing nuisance variabilities.
> 5. We removed redundant item from contribution list accordingly.
> 6. Under 9-page limitation, we tried to correct and include as many information required in the main body, moving from appendix.
> 7. We added citation of ELBO concept.
> 8. We added discussion of structure learning and Bayesian information criterion.
>
> Please do not hesitate to post further comments or questions.

---

### Official Review · AnonReviewer4 · 2020-10-28
**Interesting idea but further motivations and experimental discussions are needed.**

**Rating:** 5
**Confidence:** 3

**Review:**

Summary:
The paper presents AutoBayes: a new approach for nuisance-robust deep learning which explores different Bayesian graph models to search for the best inference strategy. It automatically builds connections between classifier, encoder, decoder, nuisance estimator and adversary DNN blocks. The approach also enables disentangling the learned representations in terms of nuisance variation and task labels. Different benchmark datasets have been used for evaluation.

##################################################################

Strenghts:
- The idea of automatically exploring various graphical models to select the best performing one is interesting.
- Overall, the paper is well written and easy to read.
- Ensemble learning is performed by stacking the explored graphical models allowing to improve performance.

##################################################################

Weaknesses:
- My main concern about the paper is that even though quantitative evaluation has been performed on several datasets with different modalities to show the benefit of using AutoBayes, the experimental evaluation section is not convincing enough as it lacks interpretation and analysis of the obtained results. For instance, one major problem addressed in this paper is models robustness to nuisance factors, however this was not discussed in this section. Hence, it would be good to include an experimental evaluation on this point.
- In Table 4 in Appendix, the simple Model B which assumes independence between Z and S performs remarkably well on task classification of the 5 first datasets since it outperforms state-of-the-art methods on all of them except QMNIST and achieves classification accuracies that are very close to the best ones (with either variational or non-variational inference), outperforming most of the presented graphical models. On the remaining datasets, Model A which is independent of S and Z, performs well compared to other graphical models especially on Faces Basics and Faces Noisy datasets. Considering these observations and perhaps the time consumption of AutoBayes, what would be the motivation of exploring all the presented graphical models (besides stacking them for ensemble learning) rather than using the simplest models A or B with good hyperparameter search? Could these results be explained by the fact that potentially some of these datasets do not present high nuisance variabilities?
- One of the main contributions as presented in Section 2 is the extensibility of the proposed framework to multiple latent representations and nuisance factors. Although this was demonstrated theoretically, it would be interesting to demonstrate this experimentally.
- One of the properties of AutoBayes is its ability to learn disentangled representations in terms of nuisance factors. In practice, how can this be evaluated?

Minor comments:
- Typo in Equation 6: xˆ  = p_μ(z1, z2) instead of xˆ  = p_μ(z1)?
- Steps to derive Equation 7 are not straightforward and can be more clarified.

---

> ### Author Response · Authors · 2020-11-24
> **Authors' Response to AnonReviewer4**
>
> We thank the reviewer for the effort spent in reviewing our paper, and for the detailed suggestions. We are happy that you found our paper interesting and well written. Although our paper is still at the proof-of-concept stage, we believe that the concept has some useful and novel insights for the community. Reflecting your valuable suggestions, we made a major revision to improve the paper. As we specifically added more experimental results and discussion to resolve your concerns, we hope you would reconsider the rating. Please find our responses below for respective queries:
> 1. On nuisance robustness: Thank you for the very important comment. We added new performance figures to facilitate the discussion of subject robustness, according to your suggestion. One of these figures shows accuracy vs. model with box-whisker plots in Fig. 9(a), where the worst/best,1st/median/3rd quartiles are present to show the dispersion of per-subject accuracies. The other figure shows accuracy vs. subject with box plots in Fig. 10, where the distribution is determined by per-model accuracies. Fig. 10 clearly shows that the performance highly depends on the subject ID. From Fig. 9(a), it can be verified that the standard classifier model A (with no consideration of nuisance $S$) is not robust against the subject variation; more specifically, the best-case user may achieve 96% accuracy whereas the worst-case user has a poor performance of 82%. The subject robustness was significantly improved by AutoBayes which explores nuisance-robust models B to Kz including A-CVAE; specifically, the worst-case user performance was improved to 94% with the explored model Fz. Besides, ensembling multiple graphical models provides non-dispersive distribution of the accuracies across different users (nuisance factors). This is an empirical evidence that the AutoBayes method is advantageous to improve the robustness against nuisance. The standard classifier, model A, has two pitfalls. On the one hand, it has limited versatility to capture task relevant features from datasets with highly dependent on nuisance factors (There is strong empirical evidence proving this point in Fig. 2. AutoBayes models perform much better on physiological datasets, such as RSVP, MI, ErrP, Faces and ASL, where inference is highly dependent on the biological state of each subject). On the other hand, the accuracy of standard classifier does not exploit much of the available information -- nuisance factors $S$ -- and therefore is unstable across different subjects. AutoBayes exploits adversarial censoring to suppress nuisance information $S$ to generate subject-invariance latent variables, and therefore has less variations across subjects as demonstrated in Fig. 9(a). We believe those additional results and discussion improved the manuscript significantly.
> 2. On hyperparameter search: Your comments are valuable. It is partly related to the comments of Reviewer2, and please refer our responses therein. In order to resolve your concern, we added a detailed comparison of the graphical models in different hyperparameter configurations (different number of hidden layers and number of nodes in each layer). From the new figure showing the trade-off between model size and accuracy in Fig. 9(b), it is evident that AutoBayes can still outperform the standard classifier model A and A-VAE model B at the same complexity for fairness comparisons. There is strong empirical evidence that exploration of various inference strategies that best fit each generative model offers significantly more contribution to model accuracy than increasing the depth of the networks of a single graphical model. Additionally, Table 4 also presents model B is sub-optimal in ASL dataset, performing only with $37.80$% accuracy. Hence, exploration of neural network topology is vastly useful as we never know which model works best and how much nuisance variabilities is inherent, given new datasets. There is also strong empirical evidence that AutoBayes models perform much better on physiological datasets, such as RSVP, MI, ErrP, Faces and ASL, where inference is highly dependent on the biological state of each subject. Leveraging the simplest models A or B on datasets that do not present high nuisance variabilities is a valid premise.
> 3. On scalability: The AutoBayes framework can be extensible to multiple latent factors. We have experimentally demonstrated it by considering zero latent models (models A,C), one latent models (models B, D-I), and two latent models (models J,K). Two-latent models perform relatively well for some datasets, while it is not always best. Of course, we could consider more nodes, but the search space will rapidly grow with the number of nodes. Please refer our responses for Reviewer2 for this scalability issues.
> 4. We made further modifications accordingly.
>
> Please do not hesitate to post further comments or questions.

---

### Official Review · AnonReviewer2 · 2020-10-28
**Enumerating all possible graphs can achieve better performance on EEG datasets, but with potential issues on scalability.**

**Rating:** 5
**Confidence:** 3

**Review:**

**Update**
I have read the author's rebuttal, and happy to see that a discussion regarding parameters is added (Figure 9). Other than that, my personal concern is similar to Anon Review 3's -- it seems that the core idea of the paper is drowned in too many technical details (granted, many of these are needed in order to implement this correctly). I wonder if a clearer discussion can be made like this -- you have a variational inference problem with certain independence assumptions, so write this out in the most abstract manner possible. To come up with a concrete objective, the question then becomes "given the factor graph, how do we add the networks (this basically lines 11 - 16 in Algorithm but not in the flow of the main text)". I think a better presentation and clarity in the main paper would greatly help acceptance.

**Overview**
The paper proposes AutoBayes, which enumerates all the plausible graphical models between data, label, and nuisance variables, remove redundant edges with d-separation rules, and learns neural networks to represent the parent-to-child information.

**Strengths**
Empirical results are very strong for certain EEG datasets such as ErrP and RSVP. It appears that on the EEG datasets, different graph structures would have very different performances so there might not be a graph structure that works equally well on all of them, which motivates the need for searching for the optimal graph structure.

**Weaknesses**

Scalability: unlike existing methods in AutoML, AutoBayes does not seem to attempt to optimize the process from which graphs are selected (i.e. pruning of graphs that are unlikely to work well), resulting in the need to enumerate all possible graphs (where the complexity is doubly-exponential with respect to the number of variables). This means that the method will have difficulty even scaling to a small amount of variables (e.g. 10).

Interpretation of the learned structures: it seems that on some datasets, CVAE already provides comparable performance to the best AutoBayes architecture, and on others best AutoBayes architecture perform much better. Can we extract any insight from the AutoBayes procedure?

Fairness of experiments: for each component of the graph, the network structure is the same; therefore, compared to a structure X->Z->Y, we have fewer parameters if we use X->Y. It is unclear as to whether the empirical improvement can be gained simply by representing edges with larger networks.

Empirical comparison: it appears that model ensembles have a significant effect over the performance. However, since one can also ensemble models from different initializations but the same architecture, it is unclear what the performance gap would become if we also do ensemble on one model but with different parameters.

**Minor suggestion**
Perhaps spend some text describing what is special about EEG datasets, and why do we expect having to predict nuisance variables to improve performance.

---

> ### Author Response · Authors · 2020-11-23
> **Authors' Response to AnonReviewer2**
>
> We thank the reviewer for the time and effort spent in reviewing our paper, and for the detailed suggestions. Your comments are all excellent in summarizing the overview and strength as well as weakness of our paper. Reflecting your valuable comments, we made a major revision adding further experimental results and discussion. We believe that our paper was significantly improved thanks to your comments. We hope you would find the usefulness of AutoBayes framework, and consider higher rating. Please find our responses below for respective queries:
> 1. On scalability: We completely agree with you that the current AutoBayes proposal has a scalability issue as it requires near-exhaustive search of models whose search space rapidly increases with the number of nodes in the graphical model. However, we would like to argue that factorizing macro nodes into a large number of micro nodes is not always useful or necessary in real-world datasets. We believe that macro-level network exploration considering only a few vertices in the Bayesian graph model is sufficient for most cases. For example, image classification tasks may have many inherent nuisance factors such as ambient light conditions, photographers' skills, camera specs, etc., and those can be represented by multiple random variables $S_1, S_2, \ldots$, instead of a single joint nuisance variable of $S$. Nevertheless, we typically have no sufficient knowledge of minor or non-dominant nuisance variations for data analysis in reality. Hence, considering only a few nuisance factors should be reasonable and realistic. Our target is to optimize subject-invariant machine learning pipelines for human-machine systems to analyze physiological signals, where we do not expect any large number of nuisance factors, other than subject identity, session number, or task conditions. We can of course model multiple factors into a single joint nuisance factor if desired. Besides nuisance nodes $S$, the scalability issue will arise when we split the latent node $Z$ into many factors $Z_1, Z_2, \ldots$. We also argue that imposing many inhomogeneous stochastic latent variables will not always be beneficial. We believe that only a few latent factors is still sufficient in practice (to explicitly represent different features). In our paper, we actually considered to have two latent variables $Z_1, Z_2$ such as model K. Its advantage over the single latent variable was not obvious. In consequence, we believe that the scalability issue will not be a major drawback for practical datasets which may not need a large number of graphical nodes. Nevertheless, we admit that the current exhaustive exploration shall be improved by more sophisticated criteria for efficient exploration. We left this challenging problem as a future work to tackle. We added some discussion on this scalability issue in the revised draft accordingly.
> 2. On interpretation of the learned structures: In our opinion, AutoBayes models perform much better on physiological datasets, such as RSVP, MI, ErrP, Faces and ASL, where inference is highly dependent on the biological state of each subject. These datasets are also event related EEG and EMG signals with notoriously low signal-to-noise ratio. It is intuitive that CVAE provides comparable performance with AutoBayes on QMNIST and Stress datasets which are less subject to variations in subject identities and have higher signal-to-noise ratio.
> 3. On fairness of experiments: Thank you for your important comment. We added a detailed comparison of the graphical models in different hyperparameter configurations (different number of hidden layers and number of nodes in each layer) to discuss the fairness. From the accuracy vs. number of model parameters plot as outlined in Fig. 9(b), it is evident that more complexity will lead to better performance while AutoBayes can still outperform the standard classifier (simple model A) at the same space complexity in higher accuracy regimes. It suggests that exploration of various inference strategies that best fit each generative model has significantly more contribution to model accuracy than increasing the depth or nodes of the neural networks of a single graphical model. More analysis of trade-off between complexity and accuracy is added in section A.10.
> 4. On empirical comparison: We tested ensemble models from different initializations but same architecture upon request by AnonReviewer2. The performance gain in the experimental results was marginal. Additionally, ensemble of even $3$ or $4$ best different graphical models is highly more efficient than the ensemble of a single graphical model with various different arrangements of hidden layers and nodes, while also performs far better in accuracy. It suggests that parallel activity of vast assemblies of different graphical models is more useful in ensembling. We added the relevant discussion in manuscript.
>
> Please do not hesitate to post further comments or questions.

---

### Decision · Program_Chairs · 2021-01-07
**Final Decision**

**Decision:**

Reject

**Comment:**

This work proposes a method for identifying appropriate graphical models through enumeration, pruning of redundant dependencies, and neural network conditionals. While structure learning is an interesting application and there are some promising results, there were a number of concerns around experimental evaluation, computational complexity of the method, clarity of the presentation, and connections to prior work. In particular, R1's concerns around the large field of structure learning in Bayesian Networks, and unwillingness to use the established terminology (and comparing to methods there) was not sufficiently addressed in the rebuttal.